# Connecting Parameter Magnitudes and Hessian Eigenspaces at Scale using Sketched Methods

**Andres Fernandez**                                              *a.fernandez@uni-tuebingen.de*
*Tübingen AI Center*
*University of Tübingen*

**Frank Schneider**                                              *f.schneider@uni-tuebingen.de*
*Tübingen AI Center*
*University of Tübingen*

**Maren Mahsereci**                                          *maren.mahsereci@uni-tuebingen.de*
*Yahoo Research*

**Philipp Hennig**                                          *philipp.hennig@uni-tuebingen.de*
*Tübingen AI Center*
*University of Tübingen*

**Reviewed on OpenReview:** *https://openreview.net/forum?id=yGGoOVpBVP*

## Abstract

Recently, it has been observed that when training a deep neural net with SGD, the majority of the loss landscape's curvature quickly concentrates in a tiny *top* eigenspace of the loss Hessian, which remains largely stable thereafter. Independently, it has been shown that successful magnitude pruning masks for deep neural nets emerge early in training and remain stable thereafter. In this work, we study these two phenomena jointly and show that they are connected: We develop a methodology to measure the similarity between arbitrary parameter masks and Hessian eigenspaces via Grassmannian metrics. We identify *overlap* as the most useful such metric due to its interpretability and stability. To compute *overlap*, we develop a matrix-free algorithm based on sketched SVDs that allows us to compute over 1000 Hessian eigenpairs for nets with over 10M parameters—an unprecedented scale by several orders of magnitude. Our experiments reveal an *overlap* between magnitude parameter masks and top Hessian eigenspaces consistently higher than chance-level, and that this effect gets accentuated for larger network sizes. This result indicates that *top Hessian eigenvectors tend to be concentrated around larger parameters*, or equivalently, that *larger parameters tend to align with directions of larger loss curvature*. Our work provides a methodology to approximate and analyze deep learning Hessians at scale, as well as a novel insight on the structure of their eigenspace.

## 1 Introduction

Deep learning (DL) benefits from overparametrization; but not all parameters are equally important. Often, a substantial portion of parameters can be *pruned*, i.e. removed, without compromising the model's performance (see Blalock et al., 2020; Hoefler et al., 2021). One efficient and popular method to identify these subnetworks is via parameter magnitude (Han et al., 2015). Interestingly, these subnetworks materialize very early in training (Frankle & Carbin, 2019), and once they emerge, their topology stops changing significantly (Achille et al., 2019; You et al., 2020). In other words, competitive subnetworks *crystallize early* in training and remain *stable* thereafter (Section 2.2, Figures 15 and 16).

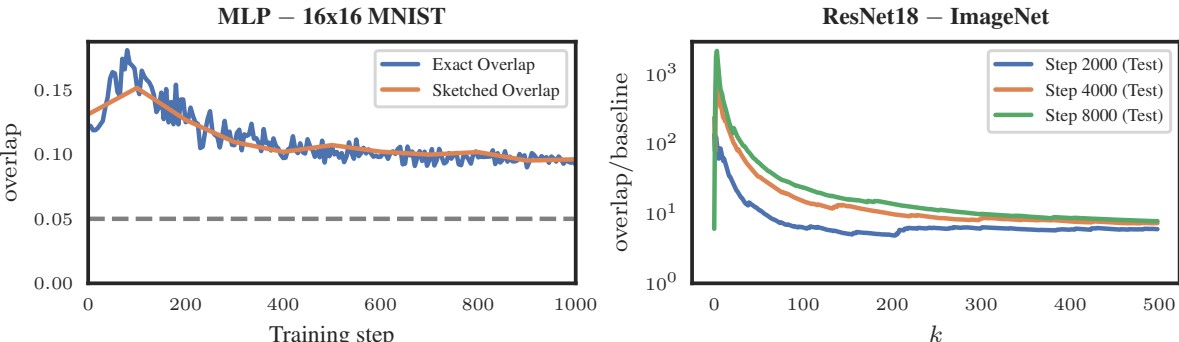

**Figure 1: Overlap between top-$k$ parameter magnitude masks and top-$k$ Hessian eigenspaces is consistently and substantially above random chance.** *(Left)* Measurements for a 7030-parameter network trained on $16 \times 16$ downsampled MNIST until convergence (see Section 6.1 and Figure 15): Exact *overlap* between top-$k$ parameters and eigenvectors with $k = 350$ (—), approximate *overlap* via sketched eigendecomposition (—), and chance-level baseline (- -) (see Section 4.2). Note how *overlap* is larger than chance, and sketched *overlap* is a good approximation. *(Right)* Lines show the ratio *sketched* overlap *vs. chance-level baseline* for a model with $>$11M parameters trained on IMAGENET (see Section 6.2), at three different points during training, and as a function of $k$. Note how *overlap* is always higher than baseline, up to a factor of 1000.

Concurrent research focuses on the loss Hessian, characterizing the loss landscape's curvature. Multiple studies found that empirically the Hessian spectrum separates into two parts: The *bulk* subspace, with many, near-zero eigenvalues, and the *top* subspace with a few eigenvalues of significantly larger magnitude (e.g. Dauphin et al., 2014; Sagun et al., 2018). Crucially, Gur-Ari et al. (2018) observed that, after initial training iterations, the gradient predominantly lies in this top subspace where it remains relatively stable throughout training. Analogous to the work on parameter masks, this indicates that the top Hessian eigenspace *crystallizes early* in training and tends to remain *stable* (Section 2.1, Figures 15 and 16).

In this work, we explore the connection between these largely independent lines of research—both reporting *early crystallization* of substructures. Our **contributions** include:

1. We establish a common theoretical ground for comparing binary parameter masks $\boldsymbol{m}_k$ (which select $k$ parameters and discard the rest) and top-$k$ eigenspaces of a Hessian. Both can be cast as rank-$k$ orthogonal matrices belonging to the same Stiefel manifold (Section 3). This allows a direct comparison of their spans using *Grassmannian metrics*[1] (see Section 4.1). We review popular Grassmannian metrics and identify *overlap* (Eq. (2)) as the most meaningful metric (Section 4.2).

2. To efficiently compute *overlap*, we develop SEIGH (Section 5 and Alg. 2), a matrix-free eigendecomposition based on sketched SVDs (Tropp et al., 2019). Our open source implementation[2] allows to compute top-$k$ Hessian eigendecompositions for $k > 10^3$ on neural networks with over 10M parameters, an unprecedented scale by orders of magnitude.

3. We provide empirical evidence that the similarities between the spaces induced by parameter magnitude masks and top-$k$ Hessian eigenspaces are consistently and substantially higher than random chance (Figure 1). This suggests that, in DL, top Hessian eigenvectors tend to be concentrated around larger parameters throughout the training process (Section 6).

Seminal work on "Optimal Brain Damage" (LeCun et al., 1989) established a theoretical link between *individual parameter values* and loss curvature, but only *at convergence*. Recent work often focuses on loss curvature for *layer-wise* parameter groups (e.g. Sankar et al., 2021), or involves just a few ($k \approx 10$) Hessian

---

[1] *Grassmannians* are manifolds of low-dimensional subspaces of a given vector space.
[2] https://github.com/andres-fr/hessian_overlap

eigenpairs without emphasis on parameters (e.g. Gur-Ari et al., 2018; Papyan, 2020; Dangel et al., 2022). In contrast, we connect Hessian *eigenspaces* to the *spans* of arbitrary parameter subsets, e.g. magnitude masks, throughout the *entire training*—including the relevant early stage—in an interpretable and scalable manner. We can thus jointly study the connection between these two phenomena and provide novel insights into a neural net's Hessian structure. This connection implies being able to approximate expensive Hessian quantities via cheap parameter inspection, which bears potential for downstream tasks where the Hessian plays a prominent role, such as optimization (Martens, 2016), pruning (LeCun et al., 1989), or uncertainty estimation (Kristiadi et al., 2021).

## 2 Hessians, Parameters and Early Crystallization

We consider a supervised classification/regression setup with train set $\mathcal{D}_{\text{train}} := \{(\boldsymbol{x}_n, \boldsymbol{y}_n)\}_{n=1}^N$ of labeled data $(\boldsymbol{x}_n, \boldsymbol{y}_n) \in \mathbb{X} \times \mathbb{Y}$, from an unknown data-generating distribution $P$. The neural net $f_{\boldsymbol{\theta}}(\boldsymbol{x}) : \mathbb{X} \to \mathbb{Y}$ maps inputs $\boldsymbol{x}$ to predictions $\hat{\boldsymbol{y}}$ via parameters $\boldsymbol{\theta} \in \mathbb{R}^D$. A loss function $\ell : \mathbb{Y} \times \mathbb{Y} \to \mathbb{R}_{\geqslant 0}$ penalizes differences between prediction $\hat{\boldsymbol{y}}$ and true label $\boldsymbol{y}$. The goal is to minimize the inaccessible *risk* $L_P(\boldsymbol{\theta}) := \int \ell(f_{\boldsymbol{\theta}}(\boldsymbol{x}), \boldsymbol{y}) dP$ via the proxy *empirical risk* $L_{\mathcal{D}_{\text{train}}}(\boldsymbol{\theta}) := \frac{1}{N} \sum_{n=1}^N \ell(f_{\boldsymbol{\theta}}(\boldsymbol{x}_n), \boldsymbol{y}_n)$. For large $N$, we approximate $L_{\mathcal{D}_{\text{train}}}(\boldsymbol{\theta})$ using *mini-batches* $\mathcal{B} \overset{\text{iid}}{\subseteq} \mathcal{D}_{\text{train}}$ of $B \ll N$ samples. For $f, \ell$ twice differentiable, we use the *gradient* $\boldsymbol{g}(\boldsymbol{\theta}) := \nabla_{\boldsymbol{\theta}} L(\boldsymbol{\theta}) \in \mathbb{R}^D$ and the *Hessian* $\boldsymbol{H}(\boldsymbol{\theta}) := \nabla_{\boldsymbol{\theta}}^2 L(\boldsymbol{\theta}) \in \mathbb{R}^{D \times D}$. With $\boldsymbol{g}, \boldsymbol{H}$ we refer to *any* loss gradient or Hessian. To emphasize the data domain, we use a subindex, e.g. $\boldsymbol{H}_{\mathcal{B}}$ for the Hessian of the mini-batch loss.

### 2.1 The Hessian in Deep Learning

The Hessian plays a prominent role in DL applications. A useful characterization of $\boldsymbol{H}$ is its eigendecomposition $\boldsymbol{H} = \boldsymbol{U} \boldsymbol{\Lambda} \boldsymbol{U}^\top = \sum_{i=1}^D \lambda_i \boldsymbol{u}_i \boldsymbol{u}_i^\top$. Here, $\boldsymbol{U}$ is orthogonal, with *eigenvectors* $\boldsymbol{u}_i$, and $\boldsymbol{\Lambda}$ is diagonal and real-valued with (ordered) *eigenvalues* $|\lambda_1| \geqslant \ldots \geqslant |\lambda_D|$. We call $\boldsymbol{U}^{(k)} := \{\boldsymbol{u}_i\}_{i=1}^k$ the *top-k eigenbasis* of $\boldsymbol{H}$, and $\text{span}(\boldsymbol{U}^{(k)})$ the *top-k eigenspace*. The top-$k$ eigendecomposition $\boldsymbol{H}^{(k)} = \sum_{i=1}^k \lambda_i \boldsymbol{u}_i \boldsymbol{u}_i^\top$ minimizes $\|\boldsymbol{H} - \boldsymbol{H}^{(k)}\|$ for all unitarily invariant norms (Golub & Van Loan, 2013, Th. 2.4.8).

Recent literature has extensively investigated the Hessian **spectrum** of neural nets, revealing that the eigenvalues are typically clustered into two parts: (1) The *bulk* of eigenvalues with near-zero magnitude and (2) a few *top* eigenvalues with significantly larger magnitude (e.g. Sagun et al., 2018; Papyan, 2019). Thus, $\boldsymbol{H}$ can be well-approximated by its top eigenpairs. For the Hessian **eigenspace**, Li et al. (2018a) showed that projecting the whole space onto a few random, fixed dimensions still allows Stochastic Gradient Descent (SGD) to perform competitively—provided enough dimensions are given—leading to the idea of an intrinsic dimensionality of problems. Gur-Ari et al. (2018) observed that this restriction to a lower-dimensional, fixed subspace seems to happen spontaneously anyway: After a brief initial training period, the gradient predominately lies within a small subspace spanned by the few top Hessian eigenvectors and this space changes little over the remaining training process. However, these phenomena might rely on optimizer and model choices (Li et al., 2018a; Ghorbani et al., 2019).

One fundamental issue greatly limiting the use of $\boldsymbol{H}$ for DL is its prohibitively large size, with $D^2$ entries. Consequently, most scalable methods are *matrix-free*, relying on Hessian-Vector Products (HVPs) to compute linear maps $\boldsymbol{w} = \boldsymbol{H}\boldsymbol{v}$ in just $\mathcal{O}(D)$ memory and time (Pearlmutter, 1994). Examples are the computation of spectral densities (Yao et al., 2020; Papyan, 2018) or top-$k$ eigendecompositions (Gur-Ari et al., 2018; Dangel et al., 2022). To make Hessian properties more accessible, specialized DL libraries have been developed recently (e.g. Dangel et al., 2020; Yao et al., 2020; Elsayed & Mahmood, 2022; Dangel et al., 2025), but efficiently accessing large portions of the Hessian remains a major challenge (see Section 5).

### 2.2 Parameter Masks and Early Crystallization

Binary parameter masks $\boldsymbol{m}_k \in \mathbb{B}^D$, with $\mathbb{B} = \{0, 1\}$, consisting of $k$ ones and the rest zeros, can be used to define subsets of parameters from a given model. A mask is $k$-sparse if it has $k$ non-zero elements. We measure mask sparsity using the ratio $\rho := k/D$. For non-binary vectors $\boldsymbol{v} \in \mathbb{R}^D$, we instead measure whether a small subset of indices $\iota$ contains a large proportion of the total norm of the vector. When $\iota$ is known, this can be directly expressed as the ratio: $\kappa(\boldsymbol{v}) := \|\boldsymbol{v}_\iota\|_2^2 / \|\boldsymbol{v}\|_2^2$ (see Hurley & Rickard, 2009). A popular choice

for $\boldsymbol{m}_k$ is to take the $k$-largest parameters by magnitude: it is computationally cheap, and has been shown to be effective for neural net pruning (Han et al., 2015), leading to smaller models that can often achieve competitive performances with $\rho \in [1\%, 10\%]$ (e.g. Gale et al., 2019; Blalock et al., 2020; Hoefler et al., 2021).

In this work, we do not perform any pruning, but rather focus on a key feature of parameter magnitude masks. Not only can they lead to competitive performance when used for pruning and be found very early in training (Frankle & Carbin, 2019; Frankle et al., 2020) but they also *stabilize soon afterwards*: You et al. (2020) compared Hamming distances between periodically extracted pruning masks and found they stop changing early in training, aligning with the loss of information plasticity reported in Achille et al. (2019). This *early crystallization* bears a striking parallel with observations made for the top Hessian subspace, and motivates our investigation: Are these two phenomena connected? How would one measure this connection? What would such a connection tell us about the parameters and the Hessian?

## 3 Parameter Masks as Orthogonal Projections

Quantifying the connection between parameter magnitude masks and top-$k$ Hessian eigenspaces—both exhibiting early crystallization—requires a way to relate a mask to a subspace. In this section, we observe that both $\boldsymbol{U}^{(k)}$ (the top-$k$ eigenbasis) and any $k$-sparse mask[3] $\boldsymbol{m}_k$ can be characterized as elements of the same compact Stiefel manifold $\mathbb{O}^{D \times k} = \{\boldsymbol{Q} : \boldsymbol{Q} \in \mathbb{R}^{D \times k}, \boldsymbol{Q}^\top \boldsymbol{Q} = \boldsymbol{I}_k\}$ (Absil et al., 2004), where $\boldsymbol{I}_k \in \mathbb{R}^{k \times k}$ denotes the identity. Specifically for masks, we further consider the subset $\mathbb{M}^{D \times k} \subset \mathbb{O}^{D \times k}$, with columns having exactly one 1, each row at most one 1, and the rest is zeros.

**Reordering parameters:** Recall from Section 2.1 that the Hessian eigenvalues are sorted by descending magnitude, exposing a single cutting point between *top* and *bulk* eigenspace at dimension $k$. To simplify notation, we impose a similar cutting point to the parameters, by defining a permutation matrix $\boldsymbol{P} \in \mathbb{M}^{D \times D}$ for any given mask $\boldsymbol{m}_k$, such that the mask entries are grouped in *selected* (i.e. $k$ entries with $m_i = 1$) and *discarded* (i.e. $m_i = 0$), i.e. $\tilde{\boldsymbol{m}} := \boldsymbol{P}^\top \boldsymbol{m} = (1, \ldots, 1, 0, \ldots, 0)$. For a parameter magnitude mask, for example, we can define $\boldsymbol{P}$ to order the parameters by nonincreasing magnitude such that $i \leqslant j \Rightarrow |\boldsymbol{Pm}|_i \geqslant |\boldsymbol{Pm}|_j$. We can now permute the parameters $\tilde{\boldsymbol{\theta}} = \boldsymbol{P}^\top \boldsymbol{\theta}$, as well as the Hessian rows and columns $\tilde{\boldsymbol{H}} := \boldsymbol{P}^\top \boldsymbol{H} \boldsymbol{P} = \tilde{\boldsymbol{U}} \boldsymbol{\Lambda} \tilde{\boldsymbol{U}}^\top$. This has no loss of generality, since $(\boldsymbol{m}, \boldsymbol{\theta}, \boldsymbol{H}) \cong (\tilde{\boldsymbol{m}}, \tilde{\boldsymbol{\theta}}, \tilde{\boldsymbol{H}})$ is an isomorphism, $\boldsymbol{H}$ and $\tilde{\boldsymbol{H}}$ are *similar*, and the loss curvature remains unaltered ($\tilde{\boldsymbol{\theta}}^\top \tilde{\boldsymbol{H}} \tilde{\boldsymbol{\theta}} = \boldsymbol{\theta}^\top \boldsymbol{H} \boldsymbol{\theta}$). Then, any permuted $k$-sparse masking operation can be expressed as:

$$\boldsymbol{P}^\top (\boldsymbol{m}_k \odot \boldsymbol{\theta}) = \tilde{\boldsymbol{m}}_k \odot \tilde{\boldsymbol{\theta}} = \underbrace{\begin{pmatrix} \boldsymbol{I}_k & 0 \\ 0 & 0 \end{pmatrix}}_{:= \tilde{\boldsymbol{\Phi}}} \tilde{\boldsymbol{\theta}} =: \boldsymbol{I}_{D,k} \boldsymbol{I}_{D,k}^\top \tilde{\boldsymbol{\theta}},$$

where $\boldsymbol{I}_{D,k} = \begin{pmatrix} \boldsymbol{I}_k \\ \boldsymbol{0} \end{pmatrix} \in \mathbb{R}^{D \times k}$, which is clearly an element of the (binary) Stiefel manifold $\mathbb{M}^{D \times k}$. Also note that this holds for any $\boldsymbol{P}$, not only for those defined for parameter magnitude masks.

**Partitioning $\tilde{\boldsymbol{H}}$:** Consider now the following partition of the reordered Hessian with $\boldsymbol{V}, \boldsymbol{D} \in \mathbb{R}^{k \times k}$, $\bar{\boldsymbol{W}}, \bar{\boldsymbol{E}} \in \mathbb{R}^{(D-k) \times (D-k)}$, and $\bar{\boldsymbol{V}}, \bar{\boldsymbol{W}}^\top \in \mathbb{R}^{(D-k) \times (k)}$:

$$\tilde{\boldsymbol{H}} := \underbrace{\left( \begin{array}{c|c} \boldsymbol{V} & \boldsymbol{W} \\ \hline \bar{\boldsymbol{V}} & \bar{\boldsymbol{W}} \end{array} \right)}_{\tilde{\boldsymbol{U}} = \boldsymbol{P}^\top \boldsymbol{U}} \underbrace{\left( \begin{array}{c|c} \boldsymbol{D} & \\ \hline & \bar{\boldsymbol{E}} \end{array} \right)}_{\boldsymbol{\Lambda}} \underbrace{\left( \begin{array}{c|c} \boldsymbol{V}^\top & \bar{\boldsymbol{V}}^\top \\ \hline \boldsymbol{W}^\top & \bar{\boldsymbol{W}}^\top \end{array} \right)}_{\tilde{\boldsymbol{U}}^\top = \boldsymbol{U}^\top \boldsymbol{P}}. \tag{1}$$

With this partition, $\tilde{\boldsymbol{U}}^{(k)} =: \begin{pmatrix} \boldsymbol{V} \\ \bar{\boldsymbol{V}} \end{pmatrix}$ is the *top-$k$* Hessian eigenbasis, and $\begin{pmatrix} \boldsymbol{W} \\ \bar{\boldsymbol{W}} \end{pmatrix}$ the *bulk* eigenbasis. Conversely, the rows of $(\boldsymbol{V}|\boldsymbol{W})$ correspond to the *selected* parameters, and $(\bar{\boldsymbol{V}}|\bar{\boldsymbol{W}})$ to the *discarded* ones. Since $\tilde{\boldsymbol{U}}$ is orthogonal, we have $\tilde{\boldsymbol{U}}^{(k)} \in \mathbb{O}^{D \times k}$, i.e. an element of the same Stiefel manifold.

## 4 Measuring Subspace Similarity via Grassmannian Metrics

We set out now to quantify the similarity between a $k$-sparse parameter mask and the top-$k$ Hessian eigenspace. Specifically, we are only interested in the similarity of their spans, since the *spaces* are the ones

---

[3]This can be generalized to masks and eigenspaces with different $k$ using Schubert varieties (Ye & Lim, 2016).

reported to undergo early crystallization. To connect parameter spaces to loss curvature, one may consider randomly perturbing the parameters, and empirically measuring the impact on the loss. This turns out to be problematic due to the non-PSD nature of $\boldsymbol{H}$ (see Appendix A.1). Various workarounds to obtain nonnegative measurements can lead to promising results, but they also entail tradeoffs (see Appendix A.2). Instead, Grassmannian metrics provide a natural and theoretically grounded way of achieving our goal, given that $k$-sparse parameter masks and the top-$k$ Hessian eigenspace can both be cast as elements of the same Stiefel manifold (Section 3). We review popular Grassmannian metrics in Section 4.1, and analyze them in more depth in Section 4.2, finding that the *overlap* metric, highlighted here, is both interpretable and stable:

$$overlap(\boldsymbol{Q}_1, \boldsymbol{Q}_2) = \frac{1}{k}\|\boldsymbol{Q}_1^\top \boldsymbol{Q}_2\|_F^2 = \frac{1}{k}\|\cos(\boldsymbol{\sigma})\|_F^2 \in [0,1] \,. \tag{2}$$

Intuitively, we can see how a zero *overlap* between $\boldsymbol{Q}_i$ and $\boldsymbol{Q}_j$ indicates that the spans of both matrices are orthogonal to each other, and an *overlap* of 1 indicates that both spaces are identical. We can also interpret *overlap* as the rotation-invariant cosine similarity between $\boldsymbol{Q}_1$ and $\boldsymbol{Q}_2$, as induced by the standard Euclidean metric on matrices.

## 4.1 Grassmann Manifolds and their Metrics

Grassmann manifolds are extensively studied (e.g. Witten, 1988; Absil et al., 2004; Bendokat et al., 2020) and have recently been applied to DL (e.g. Gur-Ari et al., 2018; Zhang et al., 2018; Dangel et al., 2022). A Grassmann manifold $\mathcal{G}_{k,D}$ is the set of all $k$-dimensional subspaces of a given $D$-dimensional Euclidean space. Two orthogonal matrices $\boldsymbol{Q}_1, \boldsymbol{Q}_2 \in \mathbb{O}^{D \times k}$ map to the same element $\mathfrak{g} \in \mathcal{G}_{k,D}$ if and only if their column span is identical. Thus, the subset of all matrices in $\mathbb{O}^{D \times k}$ that map to $\mathfrak{g}_i = \operatorname{span}(\boldsymbol{Q}_i)$ forms an *equivalence class* $\mathcal{S}_i^{\mathbb{O}} := \{\boldsymbol{Q}_j : \boldsymbol{Q}_j \boldsymbol{Z}_j = \boldsymbol{Q}_i, \ \boldsymbol{Z}_j^\top \boldsymbol{Z}_j = \boldsymbol{I}_k\}$ (e.g. Edelman et al., 1998).

The *geodesics* (i.e. shortest paths) between two elements in $\mathcal{G}_{k,D}$ are available in closed-form and can be characterized in terms of the *principal angles* $\boldsymbol{\sigma} \in [0, \frac{\pi}{2}]^k$, i.e. the "amount of rotation" required to transition from one space to another. The principal angles between two matrices in $\mathbb{O}^{D \times k}$ can be obtained via an SVD of their product, satisfying the invariance to changes within $\mathcal{S}_i^{\mathbb{O}}$ (see Appendix A.3). Based on this, there are several *Grassmannian metrics* capturing different notions of distance between subspaces (e.g. Edelman et al., 1998; Qiu et al., 2005). In the following, we highlight a few. We abbreviate $\operatorname{dist}_*(\mathfrak{g}_i, \mathfrak{g}_j) = f(\boldsymbol{\sigma}_{i \to j})$ as $\operatorname{dist}_* = f(\boldsymbol{\sigma})$, where $*$ here parametrizes any unitarily invariant norm:

a) **Geodesic distance:** This is the *arc length* of the geodesic between the respective spaces in $\mathcal{G}$, defined as $\operatorname{dist}_g = \|\boldsymbol{\sigma}\|_2 \in [0, \ \frac{\pi}{2}\sqrt{k}]$.

b) **Chordal norm:** $\operatorname{dist}_{c,*}$, obtained by minimizing $\|\boldsymbol{Q}_i \boldsymbol{Z}_i - \boldsymbol{Q}_j \boldsymbol{Z}_j\|_*$ over orthogonal matrices $(\boldsymbol{Z}_1, \boldsymbol{Z}_2)$ (for that reason it is also called *Hausdorff distance*). The $\ell_2$ and $\ell_F$ norms admit a closed-form solution in terms of principal angles: $\operatorname{dist}_{c,2} = \|2\sin\left(\frac{1}{2}\boldsymbol{\sigma}\right)\|_\infty \in [0, \sqrt{2}]$ and $\operatorname{dist}_{c,F} = \|2\sin\left(\frac{1}{2}\boldsymbol{\sigma}\right)\|_2 \in [0, \sqrt{2k}]$.

c) **Projection norm:** Also called the *gap metric*, it uses the unique orthogonal projector representation of a given subspace, i.e. $\boldsymbol{\Psi}_i = \boldsymbol{Q}_i \boldsymbol{Q}_i^\top$, as follows: $\operatorname{dist}_{p,*} = \|\boldsymbol{\Psi}_i - \boldsymbol{\Psi}_j\|_*$. Here, we also have closed-form expressions for the $\ell_2$ and $\ell_F$ norms: $\operatorname{dist}_{p,2} = \|\sin(\boldsymbol{\sigma})\|_\infty \in [0, 1]$ and $\operatorname{dist}_{p,F} = \|\sin(\boldsymbol{\sigma})\|_2 \in [0, \sqrt{k}]$.

d) **Fubini-Study:** This quantity is a measure of the *acute angle* between both spaces, generalized to higher dimensions: $\operatorname{dist}_a = \arccos\left(|\det(\boldsymbol{Q}_i^\top \boldsymbol{Q}_j)|\right) = \arccos\left(\prod_i \cos(\sigma_i)\right) \in [0, \frac{\pi}{2}]$.

e) **Overlap:** The $overlap = \frac{1}{k}\|\boldsymbol{\Psi}_i \boldsymbol{Q}_j\|_F^2 \in [0, 1]$ quantity was used in Gur-Ari et al. (2018) to measure subspace *similarity*. It is not a metric *per se*, since it is highest for equivalent subspaces and decreases with their distance, but it is a bijection of $\operatorname{dist}_{p,F}$, as follows: $\frac{1}{k}\|\boldsymbol{\Psi}_i \boldsymbol{Q}_j\|_F^2 = \frac{1}{k}\|\boldsymbol{Q}_i^\top \boldsymbol{Q}_j\|_F^2 = \frac{1}{k}\|\cos(\boldsymbol{\sigma})\|_F^2 = 1 - \|\cos(\boldsymbol{\sigma})\|_F^2 = 1 - \frac{1}{k}\operatorname{dist}_{p,F}^2$.

While the above metrics apply to any pair of matrices from $\mathbb{O}^{D \times k}$, there are also relevant metrics specific to $\mathbb{M}^{D \times k}$ (boolean subset of $\mathbb{O}^{D \times k}$ defined in Section 3), that can be characterized in a similar manner. Consider an arbitrary pair of $k$-sparse masks $(\boldsymbol{m}_i, \boldsymbol{m}_j)$, and their corresponding orthogonal projectors $\boldsymbol{\Phi} := \operatorname{diag}(\boldsymbol{m})$. Then we have:

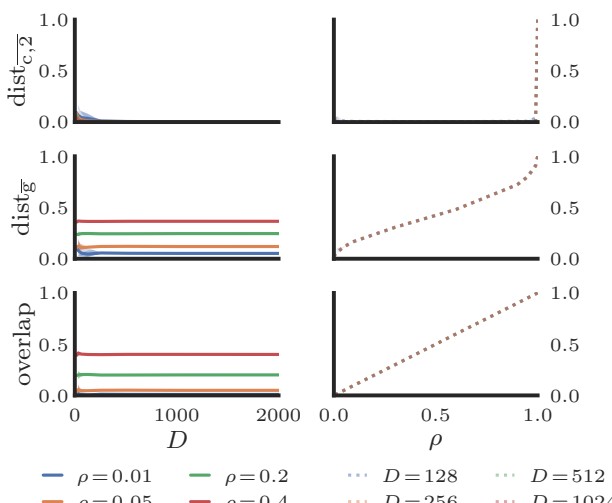

**Figure 2: Behaviour of different Grassmannian metrics for random pairs of matrices and masks.** *(Left)* As a function of $D$, for different sparsity ratios $\rho := {}^k/_D$. *(Right)* As a function of $\rho$, for different ambient dimensions $D$. Lines show the median metric for 50 random pairs, the (almost imperceptible) shaded regions span the 5-95 percentiles (see Figure 11 for broader distributions). Each row shows a selected Grassmannian metric (see Section 4.1 for more metrics): $\mathrm{dist}_{\overline{c,2}}$ is a representative example of a *collapsing* metric, being $\approx 0$ almost everywhere. The *overlap* metric is non-collapsing, and its expectation equals $\rho$.

i) **Intersection over Union (IoU):** Typically used as an evaluation metric, it is defined as the relative number of entries present in *both* masks, i.e. $\mathrm{IoU} := \frac{\boldsymbol{m}_i \cap \boldsymbol{m}_j}{\boldsymbol{m}_i \cup \boldsymbol{m}_j} \in [0, 1]$. Then we have $\boldsymbol{m}_i \cap \boldsymbol{m}_j = \|\boldsymbol{\Phi}_i \boldsymbol{\Phi}_j\|_F^2 = k \cdot overlap$, and if both masks are $k$-sparse, we also have $\boldsymbol{m}_i \cup \boldsymbol{m}_j = 2k - (\boldsymbol{m}_i \cap \boldsymbol{m}_j) = k(2 - overlap)$, yielding the bijection $overlap = \frac{2\,\mathrm{IoU}}{1 + \mathrm{IoU}}$.

ii) **Hamming distance:** This quantity, defined as the minimum number of bit-flips needed to pass from one mask to another, was used in You et al. (2020) to measure distances between pruning masks. It is in fact a Grassmannian metric: $\mathrm{dist}_{\mathrm{H}} := \|\boldsymbol{m}_i - \boldsymbol{m}_j\|_2^2 = \|\Phi_i - \Phi_j\|_F^2 = \mathrm{dist}_{p,\mathrm{F}}^2 \in [0, k]$, which means that the bijection $overlap = 1 - \frac{1}{k}\,\mathrm{dist}_{\mathrm{H}}$ also holds.

## 4.2 Comparing Grassmannian Metrics

Our aim is now to identify the most useful Grassmannian metric for comparing neural net parameter masks and Hessian eigenspaces. To empirically compare how different Grassmannian metrics change as a function of ambient dimension $D$ (for fixed $\rho$) and sparsity ratio $\rho = {}^k/_D$ (for fixed $D$), we conduct a synthetic experiment: We randomly draw matrices from $\mathbb{O}^{D \times k}$ and masks from $\mathbb{M}^{D \times k}$ and compute various Grassmannian metrics between them, normalized to be in $[0, 1]$, with higher metric indicating higher similarity. Results are highlighted in Figure 2 (experimental procedure and further experiments in Appendix A.4). This empirical analysis is complemented with a theoretical result showing that the expected *overlap* equals exactly $\rho$, as seen in Figure 2 (proof in Appendix A.5). The main insights from this analysis (Appendix A.6 extends the discussion) are:

I. For a fixed sparsity ratio $\rho$, the expectation of all metrics becomes *predictable* as $D$ increases: Already in the $D \approx 10^3$ regime (much smaller than modern DL scenarios), expectations converge tightly to values that *seem to only depend on* $\rho$ (Figure 2 left). We gathered empirical baseline values in Table 2.

II. Not all metrics are equally *informative*: Some metrics quickly collapse to 0 as $D$ increases and $\rho$ decreases (see e.g. Figure 2 (top) or Figures 12 and 13). Looking at the metric definitions in Section 4.1, we can see the reason: Collapsing metrics (e.g. $\mathrm{dist}_{\overline{c,2}}$, $\mathrm{dist}_{\overline{p,2}}$ and $\mathrm{dist}_{\overline{a}}$) are strongly influenced by individual angles, whereas non-collapsing metrics (e.g. $\mathrm{dist}_{\overline{g}}$, $\mathrm{dist}_{\overline{c,\mathrm{F}}}$, $\mathrm{dist}_{\overline{p,\mathrm{F}}}$ and *overlap*) average over all angles.

With these insights, we identify *overlap* as the most suitable metric among the reviewed ones, since: (1) *It provides a stable and efficient baseline:* Among the non-collapsing metrics, it features the smoothest expectation, equaling exactly $\rho = {}^k/_D$. This provides us with a stable, simple and theoretically-grounded baseline, establishing a **"chance-level overlap"** to compare against[4]. This is particularly beneficial for deep

---

[4]In Appendix A.7 we also compare this analytical baseline to the empirical alternative of sampling random masks, showing it yields similar results.

learning, bypassing the need to compute empirical baselines for large $D$ over many $\rho$. (2) *It is related to other metrics: overlap* can be mapped to other popular metrics such as the Hamming distance, IoU, or $\text{dist}_{p,\text{F}}$ via bijections (see Section 4.1). (3) *It has precedence in the literature:* The *overlap* metric has been used to measure eigenspace similarity in recent deep learning works (e.g. Gur-Ari et al., 2018; Dangel et al., 2022).

# 5 Sketched Hessian Eigendecompositions to Measure Overlap

To compute the *overlap* between parameter magnitude masks and top-$k$ Hessian eigenspaces, we can now simply plug in the permutation from Section 3 into Eq. (2): $\boldsymbol{Q}_1$ becomes the permuted mask matrix $\boldsymbol{I}_{D,k}$, and $\boldsymbol{Q}_2$ the correspondingly permuted top eigenspace $\tilde{\boldsymbol{U}}^{(k)}$, yielding:

$$overlap(\boldsymbol{I}_{D,k}, \tilde{\boldsymbol{U}}^{(k)}) = \frac{1}{k}\|\boldsymbol{I}_{D,k}^\top \tilde{\boldsymbol{U}}^{(k)}\|_F^2 = \frac{1}{k}\|\boldsymbol{V}\|_F^2 \,, \tag{3}$$

i.e. our problem reduces to computing the Frobenius norm of $\boldsymbol{V}$, a $k \times k$ submatrix of $\tilde{\boldsymbol{U}}^{(k)}$ (Eq. (1)).

**A case for eigendecompositions:** We aim to estimate $\|\boldsymbol{V}\|_F^2$, where $\boldsymbol{V}$ is part of $\tilde{\boldsymbol{U}}^{(k)}$, a $D \times k$ orthogonal matrix spanning the top eigenspace. Since all standard algorithms to find such matrices (Gram–Schmidt, Householder transformation, Givens rotation) require to keep track of at least $k$ full vectors (Golub & Van Loan, 2013), *a memory requirement of $\mathcal{O}(kD)$ seems inescapable in general*. Still, one may recognize that any orthogonal matrix $\hat{\boldsymbol{Q}}$ with the same span as $\tilde{\boldsymbol{U}}^{(k)}$ yields the same Frobenius norm (due to unitary invariance). However, we need to ensure that the span of said $\hat{\boldsymbol{Q}}$ does not overlap with the bulk eigenspace: It needs to be exclusively associated with the top-$k$ eigenvalues. As a consequence, *knowledge of the eigenvalues also seems inescapable in general*. In conclusion, to measure the top-$k$ *overlap*, we argue that a rank-$k$ eigendecomposition is generally needed.

**A case for *sketched* decompositions:** Among the methods that satisfy the memory requirement of $\mathcal{O}(kD)$, **orthogonal iterations** (Golub & Van Loan, 2013), an extension of Rayleigh's power method, is a popular and effective one, requiring $\mathcal{O}(k)$ measurements per iteration and $\tau$ iterations, to converge at a rate proportional to $|\lambda_{k+1}/\lambda_k|^\tau$ (Golub & Van Loan, 2013, Th. 7.3.1). This leads to large $\tau$ for smoothly decaying spectra, likely scenario for $\boldsymbol{H}$, thus $\mathcal{O}(\tau k)$ Hessian-Vector Products (HVPs) can become infeasible. Ritz accelerations provide better convergence but still suffer from this issue (Golub & Van Loan, 2013, 10.1). Alternatively, **Lanczos iterations** convergence faster (Saad, 1980), (Golub & Van Loan, 2013, 10.1.6), and can be used to approximate eigenpairs via Ritz pairs (Golub & Van Loan, 2013, 8.1.4) as well as spectral densities of e.g. large-scale deep learning Hessians via Stochastic Lanczos Quadrature (Papyan, 2018). The main issue here is that measurements must be done in sequential form, which can get very slow for $\boldsymbol{H}$, and they involve matrix powers, numerically unstable for rank-defficient matrices like $\boldsymbol{H}$. **Sketched methods** (e.g. Halko et al., 2011), based on random measurements, not only satisfy $\mathcal{O}(kD)$ memory, they also exhibit *good convergence for $\mathcal{O}(k)$ measurements* and are *parallelizable and numerically stable* (Halko et al., 2011, 1.4.2, 4.2, and 6.2). This, combined with the ability to perform matrix-free random measurements (Tropp et al., 2019, 3.2), makes it possible to compute Hessian eigendecompositions at unprecedented scales, since the main bottleneck is now the $\mathcal{O}(kD)$ memory requirement (Tropp et al., 2019, 7.1) (for example, storing 1000 eigenpairs for a ResNet-18 (He et al., 2016) with 13M float parameters requires roughly 50GB of memory, same as 100 eigenpairs for a $10\times$ larger model). Last but not least, affordable *a posteriori* methods allow to measure approximation error and rank of the recovered decomposition (Halko et al., 2011; Tropp et al., 2019).

The core idea behind the sketched SVD is that, given a linear operator $\boldsymbol{A} \in \mathbb{C}^{D_L \times D_R}$ of numerical rank $k$, any orthogonal matrices $\boldsymbol{P} \in \mathbb{C}^{D_L \times n_o}, \boldsymbol{Q} \in \mathbb{C}^{D_R \times n_o}$ that approximately capture the column and row space of $\boldsymbol{A}$, respectively, satisfy $\boldsymbol{A} \approx \boldsymbol{P}\boldsymbol{P}^*\boldsymbol{A}\boldsymbol{Q}\boldsymbol{Q}^*$ and can be efficiently obtained from $n_o = \mathcal{O}(k)$ random measurements followed by QR orthogonalization (Halko et al., 2011). Typically, $n_o \ll \min(D_L, D_R)$, which results in a *core matrix* $\boldsymbol{C} := \boldsymbol{P}^*\boldsymbol{A}\boldsymbol{Q} \in \mathbb{C}^{n_o \times n_o}$ that is much smaller[5] than $\boldsymbol{A}$ and can be decomposed via classical methods. In summary, *a potentially large and matrix-free linear operator $\boldsymbol{A}$ is decomposed into two thin matrices $(\boldsymbol{P}, \boldsymbol{Q})$ and a small matrix $\boldsymbol{C}$*. A further development features an oversampled *inner matrix* $\boldsymbol{M}_I := \boldsymbol{\Upsilon}_I^* \boldsymbol{A} \boldsymbol{\Omega}_I \in \mathbb{C}^{n_i \times n_i}$ for random measurements $\boldsymbol{\Upsilon}_I, \boldsymbol{\Omega}_I$ of columns each (Boutsidis et al., 2016), where $n_i$ is typically slightly larger

---

[5]We use lowcase $n, k$ to indicate smaller dimension than $D$.

than $n_\text{o}$. The SSVD from (Tropp et al., 2019, Sec. 2), gathered in Alg. 1, follows this idea:

$$\boldsymbol{A} \approx \boldsymbol{P}(\boldsymbol{P}^*\boldsymbol{A}\boldsymbol{Q})\boldsymbol{Q}^* \approx \boldsymbol{P} \underbrace{(\boldsymbol{\Upsilon}_I^*\boldsymbol{P})^\dagger \boldsymbol{\Upsilon}_I^*\boldsymbol{A}\boldsymbol{\Omega}_I\big[(\boldsymbol{\Omega}_I^*\boldsymbol{Q})^\dagger\big]^*}_{\boldsymbol{C}=\boldsymbol{U}\boldsymbol{\Sigma}\boldsymbol{V}^*\ \ (\text{svd})} \boldsymbol{Q}^* = (\boldsymbol{P}\boldsymbol{U})\boldsymbol{\Sigma}(\boldsymbol{V}^*\boldsymbol{Q}^*), \tag{4}$$

This yields an SVD, since $\boldsymbol{PU}$ and $\boldsymbol{QV}$ are orthogonal. It requires $n_\text{I}+2n_\text{o}$ measurements—see lines 1-3 in Alg. 1: outer measurements are used to produce $\boldsymbol{P}$ and $\boldsymbol{Q}$, and inner measurements to produce $\boldsymbol{C}$—, followed by thin matrix operations only. The memory cost is dominated by storing $\boldsymbol{P}$ and $\boldsymbol{Q}$, i.e. $\mathcal{O}(n_\text{o}(D_L+D_R))$. Arithmetic is dominated by the QR orthogonalizations needed to obtain $\boldsymbol{P}$ and $\boldsymbol{Q}$, as well as the two least-squares problems needed to solve the pseudoinverses (see (Tropp et al., 2019)). Crucially, this approximation only requires a single pass over $\boldsymbol{A}$, yields tight bounds (Tropp et al., 2019, Th. 5.1) and leads to superior performance (Tropp et al., 2019, Sec. 7), due to its numerical stability, oversampled $\boldsymbol{M}_I$, as well as uncorrelated measurements between $\boldsymbol{P}$, $\boldsymbol{Q}$ and $\boldsymbol{M}_I$ (Halko et al., 2011, 5.5) (Tropp et al., 2019, 2.8.1).

| **Algorithm 1:** SSVD (from Tropp et al. (2019)) | | **Algorithm 2:** SEIGH | |
|---|---|---|---|
| **Input:** $n_\text{I}\in\mathbb{N}$    // No. of inner measurements | | **Input:** $n_\text{I}\in\mathbb{N}$    // No. of inner measurements | |
| **Input:** $n_\text{o}\leqslant n_\text{I}$    // No. of outer measurements | | **Input:** $n_\text{o}\leqslant n_\text{I}$    // No. of outer measurements | |
| **Input:** $\boldsymbol{A}\in\mathbb{C}^{D_L\times D_R}$    // Linear operator | | **Input:** $\boldsymbol{A}\in\mathbb{C}^{D\times D}$    // Hermitian linear operator | |
| **Input:** $\boldsymbol{\Upsilon}_I\in\mathbb{C}^{D_L\times n_\text{I}}$    // Left inner random matrix | | **Input:** $\boldsymbol{\Upsilon}\in\mathbb{C}^{D\times n_\text{I}}$    // Left inner random matrix | |
| **Input:** $\boldsymbol{\Upsilon}_O\in\mathbb{C}^{D_L\times n_\text{o}}$    // Left outer random matrix | | **Input:** $\boldsymbol{\Omega}_I\in\mathbb{C}^{D\times(n_\text{I}-n_\text{o})}$    // Right inner random matrix | |
| **Input:** $\boldsymbol{\Omega}_I\in\mathbb{C}^{D_R\times n_\text{I}}$    // Right inner random matrix | | **Input:** $\boldsymbol{\Omega}_O\in\mathbb{C}^{D\times n_\text{o}}$    // Right outer random matrix | |
| **Input:** $\boldsymbol{\Omega}_O\in\mathbb{C}^{D_R\times n_\text{o}}$    // Right outer random matrix | | // $\boldsymbol{Q}\in\mathbb{C}^{D\times n_\text{o}}$, $\boldsymbol{U}\in\mathbb{C}^{n_\text{o}\times n_\text{o}}$ | |
| // $\boldsymbol{P}\in\mathbb{C}^{D_L\times n_\text{o}}$, $\boldsymbol{Q}\in\mathbb{C}^{D_R\times n_\text{o}}$, $\boldsymbol{U}\in\mathbb{C}^{n_\text{o}\times n_\text{o}}$ | | **Output:** $(\boldsymbol{Q},\boldsymbol{U},\boldsymbol{\Lambda})$ with $\boldsymbol{QU}\boldsymbol{\Lambda}\boldsymbol{U}^*\boldsymbol{Q}^*\approx\boldsymbol{A}$ | |
| **Output:** $(\boldsymbol{P},\boldsymbol{U},\boldsymbol{\Sigma},\boldsymbol{V}^*,\boldsymbol{Q}^*)$ with $\boldsymbol{PU}\boldsymbol{\Sigma}\boldsymbol{V}^*\boldsymbol{Q}^*\approx\boldsymbol{A}$ | | // Perform measurements. $[\boldsymbol{X},\boldsymbol{Y}]$ means matrix concatenation. | |
| // Perform outer measurements | | // Note the recycled $\boldsymbol{\Omega}_O$ measurements. | |
| 1   $\boldsymbol{M}_{LO}^* \leftarrow \boldsymbol{\Upsilon}_O^*\boldsymbol{A}$    // $\mathbb{C}^{n_\text{o}\times D_R}$ | | 1   $\boldsymbol{M}_O \leftarrow \boldsymbol{A}\boldsymbol{\Omega}_O$    // $\mathbb{C}^{D\times n_\text{o}}$ | |
| 2   $\boldsymbol{M}_{RO} \leftarrow \boldsymbol{A}\boldsymbol{\Omega}_O$    // $\mathbb{C}^{D_L\times n_\text{o}}$ | | 2   $\boldsymbol{M}_I \leftarrow \boldsymbol{\Upsilon}^*[\boldsymbol{A}\boldsymbol{\Omega}_I, \boldsymbol{M}_O]$   // $\boldsymbol{M}_I=\boldsymbol{\Upsilon}^*\boldsymbol{A}[\boldsymbol{\Omega}_I,\boldsymbol{\Omega}_O]\in\mathbb{C}^{n_\text{I}\times n_\text{I}}$ | |
| // Perform inner measurements | | // Orthogonalize outer measurements | |
| 3   $\boldsymbol{M}_I \leftarrow \boldsymbol{\Upsilon}_I^*\boldsymbol{A}\boldsymbol{\Omega}_I$    // $\mathbb{C}^{n_\text{I}\times n_\text{I}}$ | | 3   $(\boldsymbol{Q},\_) \leftarrow \text{qr}(\boldsymbol{M}_O)$    // $\boldsymbol{Q}$ has orthonormal columns | |
| // Orthogonalize outer measurements | | // Solve core matrix via least squares and EIGH | |
| 4   $(\boldsymbol{Q},\_) \leftarrow \text{qr}(\boldsymbol{M}_{LO})$    // $\boldsymbol{P}$ has orthonormal columns | | 4   $(\bar{\boldsymbol{U}},\bar{\boldsymbol{\Lambda}},\bar{\boldsymbol{V}}^*) \leftarrow \text{svd}(\boldsymbol{M}_I)$ | |
| 5   $(\boldsymbol{P},\_) \leftarrow \text{qr}(\boldsymbol{M}_{RO})$    // $\boldsymbol{Q}$ has orthonormal columns | | 5   $\boldsymbol{C}_L \leftarrow (\boldsymbol{\Upsilon}^*\boldsymbol{Q})^\dagger\bar{\boldsymbol{U}}$ | |
| // Solve core matrix via least squares and SVD | | 6   $\boldsymbol{C}_R \leftarrow (\boldsymbol{\Omega}^*\boldsymbol{Q})^\dagger\bar{\boldsymbol{V}}$ | |
| 6   $\boldsymbol{C} \leftarrow (\boldsymbol{\Upsilon}_I^*\boldsymbol{P})^\dagger\boldsymbol{M}_I$ | | 7   $\boldsymbol{C} \leftarrow \boldsymbol{C}_L\bar{\boldsymbol{\Lambda}}\boldsymbol{C}_R^*$    // $\boldsymbol{C}\in\mathbb{C}^{n_\text{o}\times n_\text{o}}$ is Hermitian | |
| 7   $\boldsymbol{C} \leftarrow \big[(\boldsymbol{\Omega}_I^*\boldsymbol{Q})^\dagger\boldsymbol{C}\big]^*$    // $\mathbb{C}^{n_\text{o}\times n_\text{o}}$ | | 8   $(\boldsymbol{U},\boldsymbol{\Lambda}) \leftarrow \text{eigh}(\boldsymbol{C})$    // $\boldsymbol{\Lambda}$ are the eigenvalues | |
| 8   $(\boldsymbol{U},\boldsymbol{\Sigma},\boldsymbol{V}^*) \leftarrow \text{svd}(\boldsymbol{C})$    // $\boldsymbol{\Sigma}$ are the singular values | | 9   **return** $(\boldsymbol{Q},\boldsymbol{U},\boldsymbol{\Lambda})$ | |
| 9   **return** $(\boldsymbol{P},\boldsymbol{U},\boldsymbol{\Sigma},\boldsymbol{V}^*,\boldsymbol{Q}^*)$ | | | |

**Leveraging Symmetry:** While previous works studied Hermitian extensions (Halko et al., 2011; Clarkson & Woodruff, 2017; Tropp et al., 2017), none of them proposes a single-pass sketched eigendecomposition via oversampled and uncorrelated inner matrix. In this work, we propose SEIGH (Alg. 2), a variant of SSVD that retains core oversampling while leveraging conjugate symmetry to drastically reduce the required measurements and memory. Without loss of generality (Halko et al., 2011), we assume $\boldsymbol{Q}=\boldsymbol{P}$, so only $\boldsymbol{Q}$ needs to be computed via $\boldsymbol{A}\boldsymbol{\Omega}$ outer measurements. Then, $\boldsymbol{C}$ is also Hermitian and can be eigendecomposed:

$$\boldsymbol{A} \approx \boldsymbol{Q}(\boldsymbol{Q}^*\boldsymbol{A}\boldsymbol{Q})\boldsymbol{Q}^* \approx \boldsymbol{Q}(\boldsymbol{\Upsilon}_I^*\boldsymbol{Q})^\dagger \underbrace{\boldsymbol{\Upsilon}_I^*\boldsymbol{A}\boldsymbol{\Omega}_I}_{\boldsymbol{M}_I=\bar{\boldsymbol{U}}\bar{\boldsymbol{\Lambda}}\bar{\boldsymbol{V}}^*} \big[(\boldsymbol{\Omega}_I^*\boldsymbol{Q})^\dagger\big]^*\boldsymbol{Q}^* = \boldsymbol{Q} \underbrace{(\boldsymbol{\Upsilon}_I^*\boldsymbol{Q})^\dagger\bar{\boldsymbol{U}}\bar{\boldsymbol{\Lambda}}\bar{\boldsymbol{V}}^*\big[(\boldsymbol{\Omega}_I^*\boldsymbol{Q})^\dagger\big]^*}_{\boldsymbol{C}=\boldsymbol{C}^*=\boldsymbol{U}\boldsymbol{\Lambda}\boldsymbol{U}^*\ \ (\text{eigh})}\boldsymbol{Q}^*$$

Setting $\boldsymbol{\Upsilon}_I=\boldsymbol{\Omega}_I$ would further simplify the structure, saving one pseudoinverse, but there is one caveat: We also wish to recycle part of the $\boldsymbol{A}\boldsymbol{\Omega}_I$ inner measurements to obtain $\boldsymbol{Q}$, since this is a major bottleneck for the Hessian. But recycling measurements and setting $\boldsymbol{\Upsilon}_I=\boldsymbol{\Omega}_I$ would lead to correlated measurements between $\boldsymbol{Q}$ and $\boldsymbol{M}_I$ which, as mentioned, has negative impact in the quality of this procedure (Tropp et al., 2019, 2.8.1). We opt to recycle the measurements, while keeping the uncorrelated $\boldsymbol{\Upsilon}_I=\boldsymbol{\Omega}_I$ and the two pseudoinverses.

This way, only $n_\iota$ measurements are needed, memory is roughly halved, and arithmetic is reduced by one QR decomposition and increased by one inner SVD. With SEIGH, we are now able to approximate $\|\boldsymbol{V}\|_F^2$ for large Hessians, leading to the *sketched overlap* approximation.

# 6 Magnitude Masks and Hessian Eigenspaces Overlap Substantially

We now investigate the similarity between the space spanned by the parameter magnitude masks and the top Hessian eigenspaces over the course of neural network training. We first study a small-scale toy problem (Section 6.1), where all involved quantities can still be computed exactly. This allows us to verify the existence of early crystallization, as well as the quality of our sketched *overlap* approximation. We observe an *overlap* consistently above random chance. We then move onto larger problems (up to $D > 11$M and $k = 1500$), where we also confirm that the similarity between parameter magnitude masks and top Hessian eigenspaces is orders of magnitude above random chance (Section 6.2), and that this effect is accentuated with network size. We report *overlap* results up to step 8000: since larger problems take longer to train, this covers different stages of training, but all of them reach well past initialization (see Table 1 for detailed accuracy at terminal steps). This provides us with a sufficient and convenient range to compare the observed high *overlap* across all problems.

## 6.1 Exact Computations on $16 \times 16$ MNIST

**Table 1: Overview of experimental settings**, detailing number of model parameters ($D$), learning rate ($\eta$), batch size ($B$), steps per epoch ($T$), test accuracy (acc) at step $t$, number of train/test samples used to compute $H_{train}/H_{test}$ ($N_{train}/N_{test}$ respectively), and number of SEIGH outer measurements ($n_{\mathrm{o}}$, see Alg. 2)

| Problem | Model | $D$ | $\eta$ | $B$ | $T$ | acc | $N_{train}/N_{test}$ | $n_{\mathrm{o}}$ |
|---|---|---|---|---|---|---|---|---|
| $16 \times 16$ **MNIST** | tanh-MLP Martens & Grosse (2015) | 7030 | 0.3 | 500 | 100 | 95.78% ($t = 1000$) | 500/500 | 355 |
| **CIFAR-10** | 3C3D-CNN Schneider et al. (2019) | 895,210 | 0.0226 | 128 | 312 | 74.52% ($t = 8000$) | 500/500 | 1000 |
| **CIFAR-100** | ALL-CNN-C Springenberg et al. (2015) | 1,387,108 | 0.1658 | 256 | 156 | 40.50% ($t = 8000$) | n.a./1000 | 1000 |
| **ImageNet** | RESNET-18 He et al. (2016) | 11,689,512 | 0.1 | 150 | 8207 | 17.33% ($t = 8000$) | n.a./5000 | 1500 |

For our toy problem, we seek a setup that is overparametrized enough to exhibit mask crystallization, but small enough so that full Hessian eigendecompositions can be computed. This is achieved by the setup from Martens & Grosse (2015), which is able to achieve zero training loss on downsampled MNIST digit classification with only 7030 parameters. We train using SGD as detailed in Table 1, reaching a test accuracy of 95.78% after 10 epochs (Figure 15). To compute the training and test Hessians, we use a fixed set of 500 random samples (50 per class) from each respective data split. At every fifth training step $t_i$, we compute the sparsity ratio $\kappa$ (defined in Section 2.2) for the parameters $\boldsymbol{\theta}$ and the Hessian spectrum $\boldsymbol{\Lambda}$, observing that *both subspaces experience early collapse* (Figure 15). Then, for pairs of steps $(t_i, t_j)$, we compute the IoU between successive parameter magnitude masks, as well as the overlap between successive Hessian eigenspaces. We observe that *both subspaces also experience early crystallization* (Figure 16). We reiterate that we merely *inspect* the parameter magnitude masks, but we never *apply* them, i.e., we never prune the network. Once we established our desired scenario, we compute several Grassmannian metrics between the spans of parameter magnitude masks and top Hessian eigenspaces (Figure 17). All non-collapsing metrics report that *both subspaces show a substantial and consistent similarity*. In particular, our sketched *overlap* approximation closely tracks the exact *overlap* (Figure 1, left), which is largest shortly after initialization and then decays, but still stabilizes well above random chance.

## 6.2 Sketched *overlap* on larger problems (CIFAR-10, CIFAR-100 and ImageNet)

We aim now to verify our discovery of substantial overlap in larger setups. Using the DEEPOBS framework (Schneider et al., 2019) and SGD, we train larger models on CIFAR-10/100 Krizhevsky & Hinton (2009) and on IMAGENET Deng et al. (2009) (see Table 1 for details). At steps $\{2000, 4000, 8000\}$, we gather the

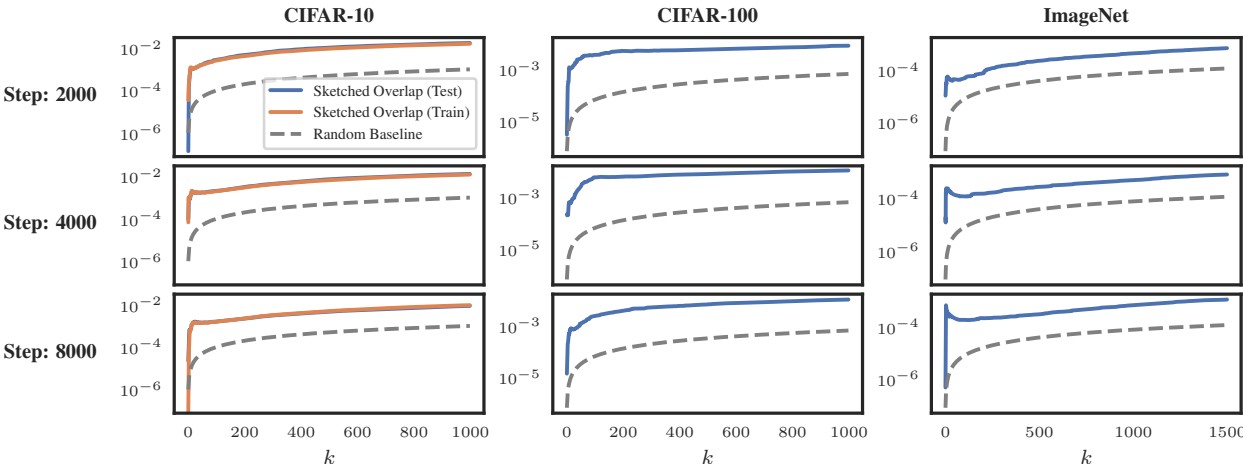

**Figure 3: Overlap between parameter magnitude masks and top Hessian eigenspaces as a function of $k$ at three different points during the training process.** Shown is the sketched *overlap* between the mask of the largest $k$ parameters by magnitude and the top-$k$ Hessian eigenspace. Consistently across the training process and for all problems and values of $k$, we observe an *overlap* substantially larger than the random chance baseline (- -) introduced in Section 4.2.

parameters and compute the top Hessian eigenspaces using SEIGH, which allows us to compute the sketched *overlap* for every $k \in \{1, ..., n_o\}$. To compute the Hessians, we choose $N_{train}/N_{test}$ to have several (balanced) samples per class, and $n_o$ to be substantially larger than the number of classes, which has been shown to be linked to the top eigenspace dimensionality (e.g. Gur-Ari et al., 2018; Papyan, 2020). We elaborate on these choices and report runtimes in Appendix B.2. Note that training CIFAR-100 and IMAGENET relies on noisy data augmentations, which lead to nondeterministic behaviour in $\boldsymbol{H}_{train}$. For this reason, we compute their *overlap* for $\boldsymbol{H}_{test}$ only. This choice is supported by the CIFAR-10 results, showing minimal difference in *overlap* for either $\boldsymbol{H}_{train}$ or $\boldsymbol{H}_{test}$.

We observe that the measured *overlap* is *significant for all problems, steps and dimensions* (Figure 3 and Figure 4, left), confirming the findings from our $16 \times 16$ MNIST toy problem. Furthermore, *this effect is accentuated by model size, surpassing a factor of $10^3$ times the baseline* for the RESNET-18 on IMAGENET (Figure 4, right). The observed sketched *overlap* also *raises in the very early steps*, perhaps linked to the crystallization covered in Section 2. We also note that, while the precision of sketched decompositions is reported to decay sharply for eigenpairs close to $n_o$ (Tropp et al., 2019, 7.9), this does not affect our observation, since the obtained *overlap* is *particularly high for the lower regimes of $k$*, far from $n_o$ and where the method is most reliable.

## 7 Conclusion

We started with the observation that, at the early stages of neural network training, both parameter magnitude masks and loss Hessian eigenspaces collapse and crystallize. To connect these two phenomena throughout the training process, we proposed a principled methodology that compares these two fundamental deep learning quantities using Grassmannian metrics, and identified *overlap* as a particularly advantageous one. We further developed SEIGH, a matrix-free sketched eigendecomposition that works for Hessians of over $10^7$ parameters, allowing us to approximate *overlap*. Our experiments reveal an *overlap* between parameter magnitude masks and top Hessian eigenspaces well above chance level for all observed problems, training steps and dimensionalities, indicating that, in DL, *top Hessian eigenvectors tend to be concentrated around larger parameters*, or equivalently, that *large parameters tend to align with directions of large loss curvature*.

While the obtained *overlap* may be considered small in absolute terms, the fact that it is orders of magnitude above chance could play a crucial role at large scales, analogously to how mini-batch gradients produce

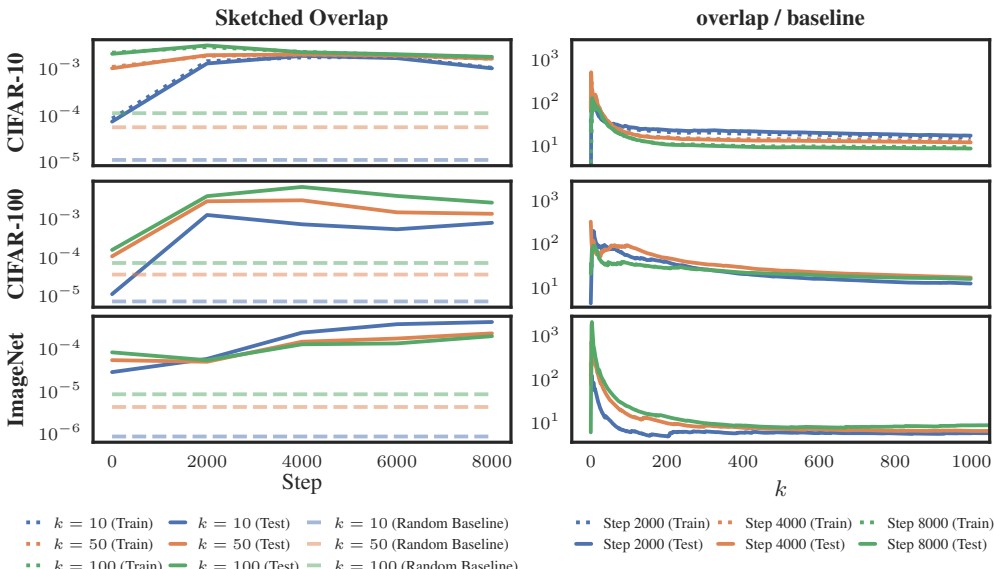

**Figure 4:** *(Left)* **Overlap between parameter magnitude masks and top Hessian eigenspaces as a function of training step.** Shown is the sketched *overlap* between masks of $k$-largest parameters by magnitude and top-$k$ Hessian eigenspaces, for different values of $k$ at different training steps. The observed *overlap* raises early on, and is consistently and substantially above random chance baseline (**- -**, introduced in Section 4.2). *(Right)* **Factor by which the measured sketched *overlap* is greater than the random baseline.** For most of the measured training process and choices of $k$, the observed *overlap* is at least $10\times$ larger than random chance baseline. This multiple factor over the random baseline is largest for small $k$ and increases with network size, surpassing $10^3$.

very noisy fluctuations in the short term, but lead to qualitative performance and generalization over larger training spans. The relevance of this connection could also bear potential for downstream tasks where the Hessian plays a prominent role (e.g. by approximating expensive Hessian quantities via cheap parameter inspection).

One main limitation of this method is that it still does not scale up to contemporary network sizes (see Appendix B.2). In this sense, we emphasize that our methodology is primarily meant as an *analysis* technique to understand and leverage the connection between parameters and loss curvature, and not necessarily part of the downstream applications themselves. In Appendix A.2, we have also explored alternative approaches that show potential as more scalable alternatives to computing *overlap*.

## Acknowledgments

The authors thank Joel A. Tropp for background on the distance between random subspaces. AF was supported by the DFG SPP 2298 (Project HE 7114/5-1). FS was supported by funds from the Cyber Valley Research Fund of the State of Baden-Württemberg. Further, the authors gratefully acknowledge financial support by the European Research Council through ERC CoG Action 101123955 ANUBIS; the DFG Cluster of Excellence "Machine Learning - New Perspectives for Science", EXC 2064/1, project number 390727645; the German Federal Ministry of Education and Research (BMBF) through the Tübingen AI Center (FKZ: 01IS18039A); and the Carl Zeiss Foundation, (project "Certification and Foundations of Safe Machine Learning Systems in Healthcare"), as well as funds from the Ministry of Science, Research and Arts of the State of Baden-Württemberg.

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

## A    Supplementary Material for Grassmannian Metrics

### A.1    Perturbing the Parameters to Measure their Importance

In order to prove that a certain subset of parameters is "sensitive" without using the theoretical framework we propose, one may consider a perturbation study where large parameters are perturbed, and the resulting loss variation is measured. More variation would then relate to more sensitivity. Here we show that, in general, such an experiment does not yield an informative quantity towards this goal (unlike our proposed Grassmannians), due to the non-PSD nature of $\boldsymbol{H}$, (i.e. negative and positive directions of curvature may cancel out for arbitrary sets of parameters) and the lack of details about the loss landscape regularity.

At any parameter vector $\boldsymbol{\theta} \in \mathbb{R}^D$, and under additive perturbations $\boldsymbol{\delta} \in \mathbb{R}^D$, the loss can be expressed via its Taylor expansion (for gradient $\boldsymbol{g}$, Hessian $\boldsymbol{H}$ and Lagrange remainder $R_3$):

$$L(\boldsymbol{\theta} + \boldsymbol{\delta}) = L(\boldsymbol{\theta}) + \boldsymbol{g}^\top \boldsymbol{\delta} + \frac{1}{2}\boldsymbol{\delta}^\top \boldsymbol{H} \boldsymbol{\delta} + R_3(\boldsymbol{\delta}) \tag{5}$$

Then, given a parameter mask $\boldsymbol{m} \in \mathbb{B}^D$, the proposed perturbation study would consist in drawing random perturbations $\boldsymbol{\delta}$ from any i.i.d. distribution $\mathcal{N}$ with zero mean $\mu$ and unit variance, and then mask said perturbations via $\boldsymbol{\delta_m} = (\boldsymbol{m} \circ \boldsymbol{\delta})$ and add them to $\boldsymbol{\theta}$ in order to estimate the expectation:

$$\mathbb{E}_{\mathcal{N}}\left[L(\boldsymbol{\theta} + \boldsymbol{\delta_m})\right] = L(\boldsymbol{\theta}) + \underbrace{\boldsymbol{g}^\top \mathbb{E}_{\mathcal{N}}\left[\boldsymbol{\delta_m}\right] + \frac{1}{2}\langle \boldsymbol{H}, \mathbb{E}_{\mathcal{N}}\left[\boldsymbol{\delta_m}\boldsymbol{\delta_m}^\top\right]\rangle + \mathbb{E}_{\mathcal{N}}\left[R_3(\boldsymbol{\delta_m})\right]}_{\varepsilon}$$

This way, we are perturbing only the selected parameters in $\boldsymbol{m}$, and measuring their impact on the (known) loss $L(\boldsymbol{\theta})$: If $|\varepsilon|$ is larger, this can be interpreted as the subset $\xi$ being more "sensitive".

Since we assumed $\boldsymbol{\mu_m} := \mathbb{E}_{\mathcal{N}}\left[\boldsymbol{\delta_m}\right] = 0$ (otherwise reparametrize $(\boldsymbol{\theta} + \boldsymbol{\delta_m}) =: (\boldsymbol{\theta} + \boldsymbol{\mu_m}) + (\boldsymbol{\delta_m} - \boldsymbol{\mu_m})$ such that the reparametrized mean of the perturbations will be 0), and we also assumed a (masked) unit covariance $\mathrm{Cov}\left[\boldsymbol{\delta_m}\right] = \boldsymbol{I_m}$, we have:

$$\mathrm{Cov}\left[\boldsymbol{\delta_m}\right] := \mathbb{E}_{\mathcal{N}}\left[(\boldsymbol{\delta_m} - \boldsymbol{\mu_m})(\boldsymbol{\delta_m} - \boldsymbol{\mu_m})^\top\right] = \mathbb{E}_{\mathcal{N}}\left[\boldsymbol{\delta_m}\boldsymbol{\delta_m}^\top\right] = \boldsymbol{I_m}$$

Then, the expectation of the perturbed loss simplifies:

$$\mathbb{E}_{\mathcal{N}}\left[L(\boldsymbol{\theta} + \boldsymbol{\delta_m})\right] = L(\boldsymbol{\theta}) + \underbrace{\boldsymbol{g}^\top \mathbb{E}_{\mathcal{N}}\left[\boldsymbol{\delta_m}\right]}_{0} + \frac{1}{2}\langle \boldsymbol{H}, \underbrace{\mathbb{E}_{\mathcal{N}}\left[\boldsymbol{\delta_m}\boldsymbol{\delta_m}^\top\right]}_{\boldsymbol{I_m}}\rangle + \mathbb{E}_{\mathcal{N}}\left[R_3(\boldsymbol{\delta_m})\right]$$

$$= L(\boldsymbol{\theta}) + \underbrace{\frac{1}{2}\mathrm{Tr}_{\boldsymbol{m}}(\boldsymbol{H}) + \mathbb{E}_{\mathcal{N}}\left[R_3(\boldsymbol{\delta_m})\right]}_{\varepsilon}$$

We see now how, in general, this technique cannot be relied upon: since $\boldsymbol{H}$ is non-PSD in general, any of its subtraces could cancel $R_3$, leading to $\varepsilon = 0$. Therefore, this procedure is not guaranteed to be informative in measuring the impact of parameter subsets on the loss curvature. In contrast, our proposed method is guaranteed to yield larger similarities whenever the selected parameters in $\boldsymbol{m}$ tend to align with directions of larger loss curvature, as shown in the paper.

Alternatively, one could consider to sample perturbations for parameters *one by one*, i.e. by setting $\boldsymbol{m}$ to equal 1 at exactly $m_i$, and 0 otherwise. Then, the expectation of the perturbed loss becomes:

$$\mathbb{E}_{\mathcal{N}}\left[L(\boldsymbol{\theta} + \boldsymbol{\delta_m})\right] = L(\boldsymbol{\theta}) + \underbrace{\frac{1}{2}\boldsymbol{H}_{ii} + \mathbb{E}_{\mathcal{N}}\left[R_3(\boldsymbol{\delta_m})\right]}_{\varepsilon_i} \tag{6}$$

The idea here is that the aggregated perturbation $\varepsilon' = \sum_i |\varepsilon_i|$ is now more likely to be informative, since positive and negative diagonal entries don't cancel anymore. This still entails a problematic trade-off that favours our Grassmannian characterization, for the following reasons: (1) In DL, the loss landscape may be far from convex or regular (e.g. Li et al., 2018b, Fig. 1), and to interpret $\varepsilon'$ meaningfully we would still require some characterization of $R_3$ similar in spirit to the theoretically founded Grasmannian that we use. (2) This method requires to estimate one expectation per scalar parameter. For a set of $k$ parameters and $n$ samples per expectation, this means performing $\mathcal{O}(n \cdot k)$ forward propagations over a given dataset. In contrast, our method is theoretically grounded and our proposed sketched method requires just $\mathcal{O}(k)$ HVPs (see Section 5 and Halko et al. (2011, 4.2) for more discussion).

## A.2 Beyond Perturbing the Parameters to Measure their Importance

Despite the various sources of error and computational issues related to parameter perturbations (see Appendix A.1), one may still feel compelled to explore related strategies, seeking a more efficient and conceptually simple alternative to our proposed Grassmannian methodology. In this section, we introduce three such alternatives which, despite of some issues, constitute a promising line of investigation. We discuss here some of their main aspects in terms of model error, computation and empirical behaviour.

**Nonnegative Gauss-Newton Subtraces:** A main issue with $\boldsymbol{H}$ is its non-PSD nature. Instead, the idea here is to target the Gauss-Newton matrix $\boldsymbol{G}$, which is a PSD approximation to $\boldsymbol{H}$ (Schraudolph, 2002).

Since PSD matrices have nonnegative diagonal entries, $\boldsymbol{G}_{ii}$ could serve as a proxy measurement, denoting parameter sensitivity for $\boldsymbol{\theta}_i$. In that case, a larger trace for a given subset of indices could mean that the corresponding parameter space is associated with larger loss curvature.

Instead of casting this as a parameter perturbation study, which would be exposed to the $R_3$ approximation error as well as the impossibility of disentangling $\boldsymbol{G}$ from $\boldsymbol{H}$ (see Appendix A.1), we propose here to *measure the diagonal entries of $\boldsymbol{G}$ one by one using Gauss-Newton Vector Products (GNVPs)*. Since we are just interested in the entries corresponding to the $k$ largest parameters by magnitude, this just requires $\mathcal{O}(D)$ memory and $\mathcal{O}(k)$ GNVPs, thus being fully parallelizable and substantially faster than computing *overlap*. Furthermore, GNVPs are roughly twice as fast as HVPs, and require half the memory (Martens, 2010, 4.2).

Once we have obtained the top-$k$ diagonal entries (i.e. the diagonal entries of $\boldsymbol{G}$ corresponding to the $k$-largest parameters), we need a way to compare them with *overlap*. For this, we propose to compute the following *normalized trace summation*:

$$\xi_k := \frac{1}{\text{Tr}(\boldsymbol{G})} \sum_{i \in \{i_1 \dots i_k\}} \boldsymbol{G}_{ii} \quad \in [0, 1] \tag{7}$$

Where the $i_1, \dots, i_k$ indices run over the parameters by descending magnitude. We can see that $\xi$ increases monotonically with $k$, reaching $\xi_D = 1$. Furthermore, we define the baseline corresponding to a uniformly distributed trace as $\bar{\xi}_k := k/D$. This baseline behaves analogously to our *"chance-level" overlap* baseline from Section 4.2: if $\xi_k > \bar{\xi}_k$, the top-$k$ parameters are associated with an above-chance trace (which, as already mentioned, we treat as a proxy for curvature in the case of $\boldsymbol{G}$). See Figures 6 and 7 for an illustration.

**Nonnegative Squared-Hessian Subtraces:** The rationale and computations involved here are identical as for the previously described $\boldsymbol{G}$, but using $\boldsymbol{H}^2$ instead. This is viable, since $\boldsymbol{H}^2$ is also PSD and $[\boldsymbol{H}^2]_{ii}$ can be efficiently and exactly computed with a single HVP as follows:

$$[\boldsymbol{H}^2]_{ii} = \boldsymbol{e}_i^\top \boldsymbol{H}^2 \boldsymbol{e}_i = \|\boldsymbol{H}\boldsymbol{e}_i\|_2^2$$

This technique is also substantially faster than *overlap*, requiring only $\mathcal{O}(D)$ memory and $\mathcal{O}(k)$ parallelizable HVPs. Although we do not assume $\boldsymbol{H}^2 \approx \boldsymbol{H}$, this experiment aims to explore whether subtraces of $\boldsymbol{H}^2$ can also be treated as proxy for curvature, in the same fashion as previously discussed for $\boldsymbol{G}$.

**Adversarial Perturbations:** Recall from Appendix A.1 and Eq. (6) that a random perturbation study would require $\mathcal{O}(n \cdot k)$ forward propoagations, for potentially large $n$. Instead of random sampling, the idea here is to *use a single "optimal" perturbation, such that curvature is maximized in that direction*. This is precisely the idea behind Sharpness-Aware Minimization (SAM) (Foret et al., 2021), where the perturbation with constrained radius $\lambda \in \mathbb{R}_+$ such that the loss $L(\boldsymbol{\theta} + \lambda \boldsymbol{\epsilon}^*)$ is maximized, has the approximate closed-form solution $\boldsymbol{\epsilon}^* = \boldsymbol{g}/\|\boldsymbol{g}\|_2$. In our setting from Eq. (6), we want to evaluate the following *masked loss delta*:

$$\Delta_{\lambda,i} = |L(\boldsymbol{\theta}) - L(\boldsymbol{\theta} + \lambda(\boldsymbol{e}_i \circ \boldsymbol{\epsilon}^*))|$$

where the $i$ index covers each of the top-$k$ parameters by magnitude. This doesn't rely on PSD-ness anymore, since $\Delta$ comprises absolute values for individual parameters. It is also very fast to compute, since we just need $\mathcal{O}(D)$ memory, one forward- and backpropagation to obtain $L(\boldsymbol{\theta})$ and $\epsilon$, plus one additional forward propagation per $i$. Once we have the *deltas*, we can define a normalized summation, analogous to Eq. (7), that can be used to compare against $\bar{\xi}_k$ (see Figures 6 and 7):

$$\zeta_{\lambda,k} =:= \frac{1}{\sum_{i=1}^D \Delta_{\lambda,i}} \sum_{i \in \{i_1 \dots i_k\}} \Delta_{\lambda,i} \quad \in [0, 1] \tag{8}$$

**Experimental results:** So far, we introduced three measurement techniques that are more efficient than *overlap*. We now ask: *Can they be used as a reliable proxy for parameter sensitivity? Do they yield measurements above baseline? How do they relate to overlap?* To answer this, we gathered the above metrics for steps 0, 100 and 1000 of the MNIST setup described in Section 6.1. The steps were chosen to cover initialization, mid-training and convergence, as illustrated in Figure 5. Results can be found in Figures 6 and 7, together with the corresponding *overlap* and uniform baseline $\bar{\xi}$ for comparison. We chose 3 different values of $\lambda$ to cover different perturbation scales.

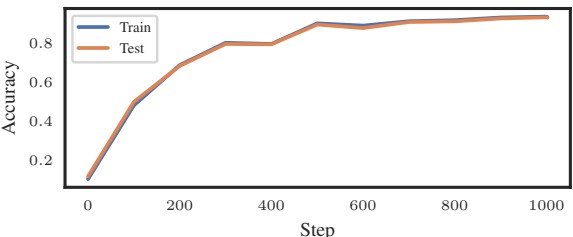

**Figure 5:** Training evolution of the model used to report Figures 6, 7 and 14. It corresponds to the 7030-parameter model trained on $16 \times 16$ downsampled MNIST, as described in Section 6.1.

Comparing both figures, we first observe that training and test data look quite similar throughout the whole training, which is consistent with the small gap between training and test accuracies from Figure 5. Crucially, we observe that, like *overlap*, *all 3 proxy measurements tend to be well above chance-level baseline in most of their domain*, supporting their potential use as scalable replacements for *overlap*, and suggesting a possible connection to the Hessian eigenspace. This is promising, but it comes with the caveat of *more unstable behaviour, both across k*—all proxies tend to be below baseline for small $k$—, *and across training step*—they can go well above *overlap* early on, but then they steeply collapse towards baseline. The instability of these overlap proxies, together with their approximate nature, makes them a challenging but potentially rewarding target for future research.

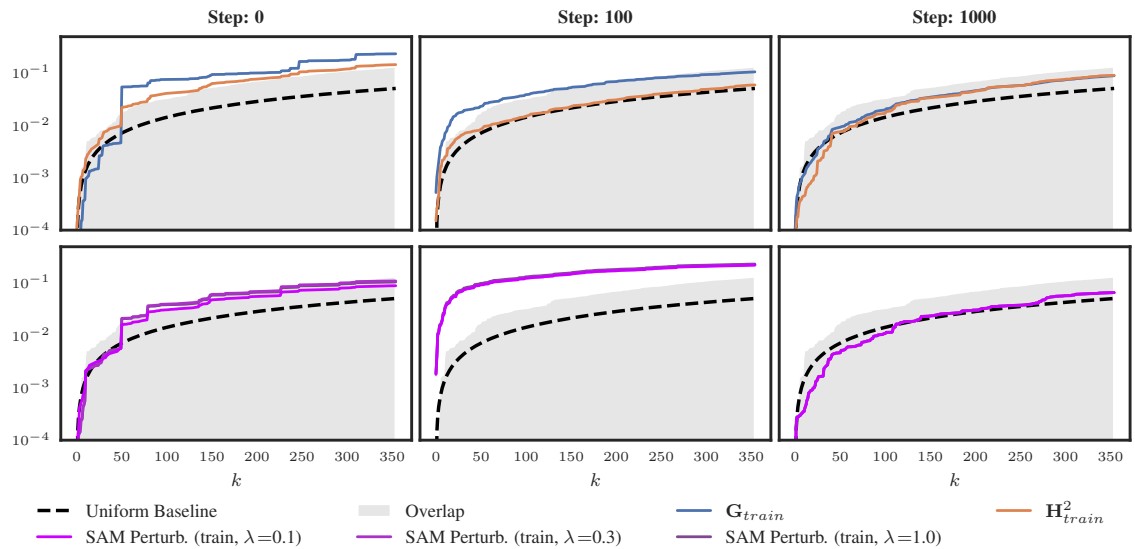

**Figure 6: Proxy *overlap* measurements, training split:** *(top row)* The $\boldsymbol{G}$ *normalized trace summation* feature from Eq. (7), as well as the corresponding $\boldsymbol{H}^2$ normalized trace summation. *(bottom row)* The SAM perturbation feature from Eq. (8), obtained for three different values of $\lambda$, to cover different perturbation scales. *(both rows)* All features are compared to the uniform baseline $k/D$ and the *overlap* associated to the $k$-largest parameters by magnitude. See Figure 5 for respective accuracies at given steps.

## A.3 Main Properties of Grassmannian Metrics

A desirable property of Grassmann manifolds is the availability of closed-form expressions for their *geodesics* (i.e. the shortest paths between any two elements $\mathfrak{g}_i, \mathfrak{g}_j \in \mathcal{G}$) and metric functions. We can thus measure distances between subspaces in an interpretable and computationally amenable manner: geodesics from $\mathfrak{g}_i$ to

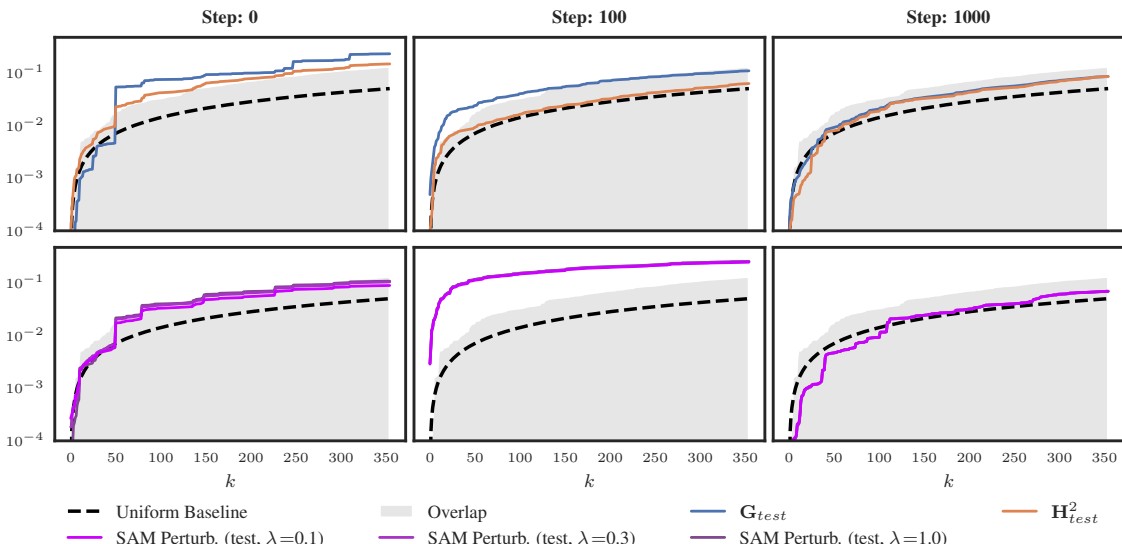

**Figure 7: Proxy *overlap* measurements, test split:** *(top row)* The $\boldsymbol{G}$ *normalized trace summation* feature from Eq. (7), as well as the corresponding $\boldsymbol{H}^2$ normalized trace summation. *(bottom row)* The SAM perturbation feature from Eq. (8), obtained for three different values of $\lambda$, to cover different perturbation scales. *(both rows)* All features are compared to the uniform baseline $k/D$ and the *overlap* associated to the $k$-largest parameters by magnitude. See Figure 5 for respective accuracies at given steps.

$\mathfrak{g}_j$ follow circular trajectories, and therefore their *distance* can be interpreted as the "amount of rotation" needed to go from one space to another. Such rotations can be succinctly expressed in terms of *principal angles* $\boldsymbol{\sigma}_{i \to j} \in [0, \frac{\pi}{2}]^k$, and they can be efficiently obtained from $\boldsymbol{Q} \in \mathbb{O}^{D \times k}$ matrices via their Singular Value Decomposition (SVD) (e.g. Edelman et al., 1998, s. 4.3):

$$\boldsymbol{Q}_i^\top \boldsymbol{Q}_j =: \boldsymbol{L}_{i \to j} \operatorname{diag}\left(\cos(\boldsymbol{\sigma}_{i \to j})\right) \boldsymbol{R}_{i \to j}^\top, \quad \boldsymbol{L}, \boldsymbol{R} \text{ orthogonal.} \tag{9}$$

Since $\boldsymbol{Q}_i, \boldsymbol{Q}_j$ have orthonormal columns, $\cos(\boldsymbol{\sigma}_{i \to j}) \in [0,1]^k$ (e.g. Neretin, 2001), and more similar spans will yield larger singular values, which translate to smaller rotations. Importantly, singular values are invariant under similarity:

$$\operatorname{diag}\left(\cos(\boldsymbol{\sigma}_{i \to j})\right) = \boldsymbol{L}_{i \to j}^\top \boldsymbol{Q}_i^\top \boldsymbol{Q}_j \boldsymbol{R}_{i \to j} \tag{10}$$

$$= \boldsymbol{L}_{i \to j}'^\top (\boldsymbol{Z}_i^\top \boldsymbol{Q}_i^\top)(\boldsymbol{Q}_j \boldsymbol{Z}_j) \boldsymbol{R}_{i \to j}'. \tag{11}$$

In other words, they are invariant under the action of any orthogonal $\boldsymbol{Z}$, which means that $\boldsymbol{\sigma}$ does not change if we replace an input matrix with any other matrix from the same equivalence class (see definition for $\mathcal{S}_i^{\mathbb{O}}$ in Section 4.1). The family of functions that satisfy this invariance, plus the axioms of metric spaces, form the family of *Grassmannian metrics* (e.g. Qiu et al., 2005), each capturing a different notion of distance between subspaces (largest principal angle, sum of principal angles…). See Section 4.1 for more details on specific metrics.

## A.4 Synthetic Experiment on Grassmannian Metrics

Here we provide details about the synthetic experiment discussed in Section 4.2. We compute the reviewed Grassmannian metrics **a)** to **e)** from Section 4.1 between randomly drawn matrices from $\mathbb{O}^{D \times k}$ and masks from $\mathbb{M}^{D \times k}$. We use the subgroup algorithm (Diaconis & Shahshahani, 1987) to sample matrices uniformly from $\mathbb{O}^{D \times k}$ and column permutations of $\boldsymbol{I}_{D,k}$ to sample uniformly from $\mathbb{M}^{D \times k}$. To determine if the metrics behave differently for masks than for general orthogonal matrices, we inspect three different *modalities*: pairs of matrices ($\mathbb{O}$-to-$\mathbb{O}$, Figures 8 and 9), masks ($\mathbb{M}$-to-$\mathbb{M}$, Figures 10 and 11), and matrix-mask pairs ($\mathbb{O}$-to-$\mathbb{M}$, Figures 12 and 13). We normalize all metrics, denoted by $\operatorname{dist}_{\overline{\ast}}$, to be in $[0, 1]$, with 1 indicating highest similarity (i.e. smallest distance). The overall procedure is gathered in Alg. 3.

---

**Algorithm 3:** Synthetic experiment on Grassmannian metrics (see Section 4.2 for details).

**Input:** $\{D_1, D_2, ...\}$      `// Matrix height (`$D_i \in \mathbb{N}$`)`
**Input:** $\{r_1, r_2, ...\}$      `// Width-to-height ratio (`$r_i \in [0,1]$`)`
**Input:** $\{(\mathbb{O}, \mathbb{O}), (\mathbb{O}, \mathbb{M}), (\mathbb{M}, \mathbb{M})\}$      `// Modality`
**Input:** $\{\text{dist}_{\overline{g}}, \text{dist}_{\overline{c,2}}, \text{dist}_{\overline{c,F}}, \text{dist}_{\overline{p,2}}, \text{dist}_{\overline{p,F}}, \text{dist}_{\bar{a}}, overlap\}$      `// Metric (normalized)`
**Input:** $T$      `// Number of random samples`

1   $\mathcal{R} \leftarrow \varnothing$      `// Result (a dictionary)`
2   **for** $d \in \{D_1, D_2, ...\}$ **do**
3     **for** $r \in \{r_1, r_2, ...\}$ **do**
4       $k \leftarrow \max(1, \text{round}(r \cdot d))$
5       **for** $\text{dist} \in \{\text{dist}_{\overline{g}}, \text{dist}_{\overline{c,2}}, \text{dist}_{\overline{c,F}}, \text{dist}_{\overline{p,2}}, \text{dist}_{\overline{p,F}}, \text{dist}_{\bar{a}}, overlap\}$ **do**
6         **for** $(\mathcal{M}_1, \mathcal{M}_2) \in \{(\mathbb{O}, \mathbb{O}), (\mathbb{O}, \mathbb{M}), (\mathbb{M}, \mathbb{M})\}$ **do**
7           $\mathcal{H} \leftarrow \varnothing$      `// Collection of samples`
8           **for** $\{1, ..., T\}$ **do**
9             $(\boldsymbol{Q}_1, \boldsymbol{Q}_2) \overset{\text{unif.}}{\sim} (\mathcal{M}_1^{d \times k}, \mathcal{M}_2^{d \times k})$ $\mathcal{H} \leftarrow \mathcal{H} \cup \text{dist}\big(\text{span}(\boldsymbol{Q}_1), \text{span}(\boldsymbol{Q}_2)\big)$
10           **end**
11           $\mathcal{R}_{[d,r,\text{dist},\mathcal{M}_1,\mathcal{M}_2]} \leftarrow \mathcal{H}$      `// Gather samples into result`
12         **end**
13       **end**
14     **end**
15   **end**
16   **return** $\mathcal{R}$

---

We want to inspect how the distribution of the metrics changes as a function of sparsity $k$ and sparsity ratio $\rho = \frac{k}{D}$. First, given fixed values of $\rho$, we study the distribution of all Grassmannian metrics as a function of $D$ (Figures 8, 10 and 12). Additionally, given fixed values of $D$ we study how the Grassmannian metrics change as a function of $\rho$ (Figures 9, 11 and 13). We used the following values:

- Number of random (matrix or mask) samples: $T = 50$

- For Figures 8, 10 and 12 we investigate four different (fixed) ratios $\rho \in \{0.4, 0.2, 0.05, 0.01\}$ at several increasing dimensions $d \in \{16, 32, 64, 128, 256, 512, 1024, 2048\}$.

- For Figures 9, 11 and 13 we investigated four (fixed) dimensions $d \in \{128, 256, 512, 1024\}$ at several increasing ratios $\rho \in \{0.005, 0.01, 0.05, 0.1, 0.33, 0.66, 0.9, 0.95, 0.99, 1\}$.

Note how, in the second case, we concentrate the sparsity ratios around the extremes. This is to better capture the behaviour of collapsing metrics, as discussed in Section 4.2.

## A.5 Expectation of *overlap* for Uniformly Random Matrices

In Section 4.2 we empirically observed that the expectation under uniformly distributed random matrices becomes *predictable* for all reviewed metrics, converging to a baseline value. Here we show that, for *overlap*, such expectation equals exactly $\rho = \frac{k}{D}$. This lemma depends on a standard calculation that was communicated to us by Joel A. Tropp.

**Lemma A.1**
*Let $\boldsymbol{Q}_1, \boldsymbol{Q}_2$ be random matrices drawn uniformly from the Stiefel manifold $\mathbb{O}^{D \times k} := \{\boldsymbol{Q} : \boldsymbol{Q} \in \mathbb{R}^{D \times k}, \boldsymbol{Q}^\top \boldsymbol{Q} = \boldsymbol{I}_k\}$. Then,*

$$\mathbb{E}\left[\text{overlap}(\text{span}(Q_1), \text{span}(Q_2))\right] = \frac{k}{D} \tag{12}$$

*Proof.* We start by rewriting the definition of *overlap*, presented in Section 4.1, in terms of the trace of orthogonal projectors $\boldsymbol{\Psi} = \boldsymbol{Q}\boldsymbol{Q}^\top \in \mathbb{R}^{D \times D}$. We make use of the idempotence of $\boldsymbol{\Psi}$ and the unitary invariance

of the Frobenius norm:

$$overlap(\text{span}(\boldsymbol{Q}_1), \text{span}(\boldsymbol{Q}_2)) = \frac{1}{k}\|\boldsymbol{Q}_1^\top \boldsymbol{Q}_2\|_F^2 = \frac{1}{k}\|\boldsymbol{\Psi}_1 \boldsymbol{\Psi}_2\|_F^2 = \frac{1}{k}\text{Tr}(\boldsymbol{\Psi}_1 \boldsymbol{\Psi}_2^2 \boldsymbol{\Psi}_1) \tag{13}$$

$$= \frac{1}{k}\text{Tr}(\boldsymbol{\Psi}_1 \boldsymbol{\Psi}_2 \boldsymbol{\Psi}_1) \tag{14}$$

We further observe that, if $\boldsymbol{Q}$ is drawn uniformly from $\mathbb{O}^{D \times k}$, the marginal distribution of every column $\boldsymbol{q}$ is the uniform distribution over the Euclidean unit sphere, hence it is isotropic:

$$\mathbb{E}\left[\boldsymbol{q}\boldsymbol{q}^\top\right] = \frac{1}{D}\boldsymbol{I}_D \tag{15}$$

Where $I_D$ is the rank-$D$ identity matrix. Then, leveraging linearity of expectation, the orthogonal projector $\boldsymbol{\Psi} = \boldsymbol{Q}\boldsymbol{Q}^\top$ can be expressed as follows:

$$\mathbb{E}\left[\boldsymbol{\Psi}\right] = \sum_{i=1}^{k}\mathbb{E}\left[\boldsymbol{q}_i\boldsymbol{q}_i^\top\right] = \frac{k}{D}\boldsymbol{I}_D \tag{16}$$

Now, given two independent realizations $\boldsymbol{Q}_1, \boldsymbol{Q}_2$, we form the associated orthogonal projectors $\boldsymbol{\Psi}_1, \boldsymbol{\Psi}_2$. Write $\mathbb{E}_1, \mathbb{E}_2$ for the expectations of the respective distributions of $\boldsymbol{Q}_1$ and $\boldsymbol{Q}_2$. Then, leveraging independence of $\boldsymbol{Q}_1$ and $\boldsymbol{Q}_2$, idempotence of $\boldsymbol{\Psi}$, linearity of expectation, and $\text{Tr}(\boldsymbol{\Psi}) = k$, we have:

$$\mathbb{E}\left[\text{Tr}(\boldsymbol{\Psi}_1 \boldsymbol{\Psi}_2 \boldsymbol{\Psi}_1)\right] = \mathbb{E}_1\left[\text{Tr}(\boldsymbol{\Psi}_1 \mathbb{E}_2\left[\boldsymbol{\Psi}_2\right] \boldsymbol{\Psi}_1)\right] = \frac{k}{D}\mathbb{E}_1\left[\text{Tr}(\boldsymbol{\Psi}_1 \boldsymbol{I}_D \boldsymbol{\Psi}_1)\right] = \frac{k}{D}\text{Tr}(\boldsymbol{\Psi}_1) = \frac{k^2}{D} \tag{17}$$

Replacing in the definition of *overlap* concludes the proof. $\square$

**Table 2: Empirical baselines for Grassmannian metrics.** Shown are the measured expectations (averaged over 50 samples at $D = 2048$) of different Grassmannian metrics between two uniformly random matrices in $\mathbb{O}$ for different values of $\rho = \frac{k}{D}$. Note how *overlap* converges to exactly $\rho$ (see Appendix A.5).

| Metric | $\rho$ | | | | |
|---|---|---|---|---|---|
| | 0.005 | 0.01 | 0.05 | 0.2 | 0.4 |
| $\text{dist}_{\overline{g}}$ | 0.03711 | 0.05191 | 0.11909 | 0.24534 | 0.36536 |
| $\text{dist}_{\overline{c,2}}$ | 0.00197 | 0.00161 | 0.00048 | 0.00046 | 0.00026 |
| $\text{dist}_{\overline{c,\text{F}}}$ | 0.02983 | 0.04202 | 0.09984 | 0.21754 | 0.33783 |
| $\text{dist}_{\overline{p,2}}$ | $10^{-5}$ | $10^{-5}$ | 0 | 0 | 0 |
| $\text{dist}_{\overline{p,\text{F}}}$ | 0.00248 | 0.00479 | 0.02513 | 0.1057 | 0.22527 |
| $\text{dist}_{\overline{a}}$ | 0 | 0 | 0 | 0 | 0 |
| *overlap* | 0.00495 | 0.00955 | 0.04962 | 0.20022 | 0.39980 |

## A.6 Extended Discussion on Grassmannian Metrics

We extend here the main insights presented in Section 4.2:

**Mask-vs-mask metrics have larger variance, and in some cases lower expectations:** All distributions for the mask-vs-mask modality have higher variance compared to the other modalities (see Appendix A.4). Furthermore, the distributions for the $\text{dist}_{\overline{g}}$ and $\text{dist}_{\overline{c,\text{F}}}$ metrics seem to follow lower trajectories for the mask-vs-mask modality, compared to the other modalities. This is not the case for $\text{dist}_{\overline{p,\text{F}}}$ and *overlap*, whose expectation does not seem to be affected by the modality.

**Extremal values of $\rho$ lead to saturation, except for *overlap*:** Most metrics exhibit a nonlinear behaviour near the extremes (Figure 9). This is particularly so for collapsing metrics, but saturation can also be observed in the non-collapsing ones. The only exception is *overlap*, whose expectation is linear as shown in Appendix A.5.

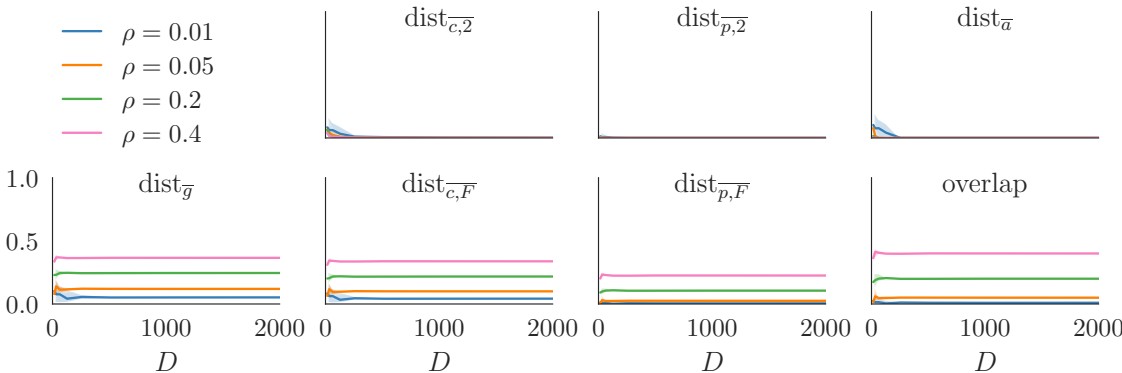

**Figure 8: Grassmannian metrics for random pairs of matrices** $(\boldsymbol{Q}_1, \boldsymbol{Q}_2) \overset{\text{unif.}}{\sim} (\mathbb{O}^{D \times k}, \mathbb{O}^{D \times k})$ **as a function of** $D$**.** Each subplot shows a different metric, and each color corresponds to a different ratio $r = \frac{k}{D}$. For each $(D, k)$, we sample 50 random pairs and report the median of the resulting distribution (line plot) as well as the 5-95 percentiles (shaded regions).

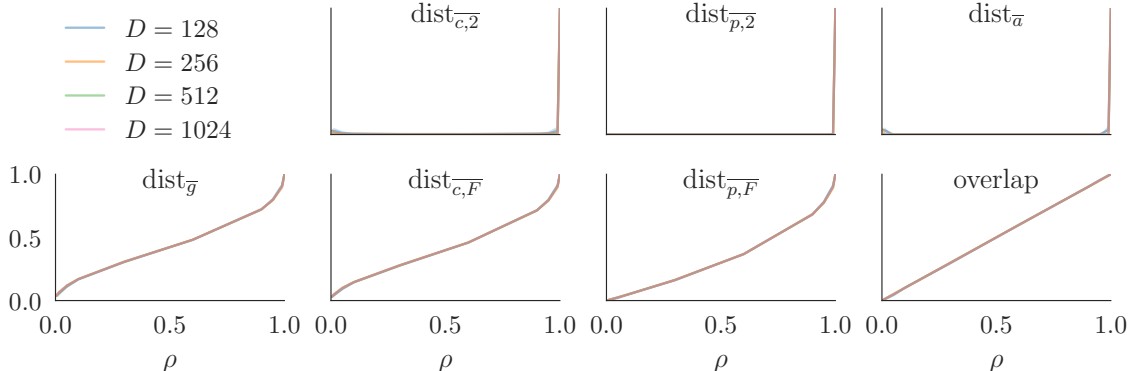

**Figure 9: Grassmannian metrics for random pairs of matrices** $(\boldsymbol{Q}_1, \boldsymbol{Q}_2) \overset{\text{unif.}}{\sim} (\mathbb{O}^{D \times k}, \mathbb{O}^{D \times k})$ **as a function of** $\rho = \frac{k}{D}$**.** Each subplot shows a different metric, and each color corresponds to a different dimension $D$. For each $(D, k)$, we sample 50 random pairs and report the median of the resulting distribution (line plot) as well as the 5-95 percentiles (shaded regions).

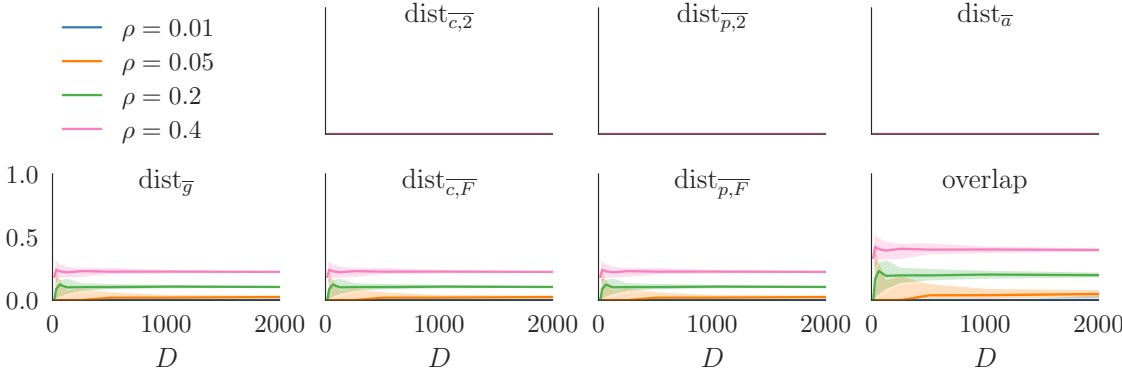

**Figure 10: Grassmannian metrics for random pairs of masks** $(\boldsymbol{Q}_1, \boldsymbol{Q}_2) \overset{\text{unif.}}{\sim} (\mathbb{M}^{D \times k}, \mathbb{M}^{D \times k})$ **as a function of** $D$ **and for fixed** $\rho$**.** Each subplot shows a different Grassmannian metric with the different lines indicating four different ratios $\rho$. For each value of $D$, we report the median metric over 50 random pairs with the shaded regions showing the 5-95 percentiles.

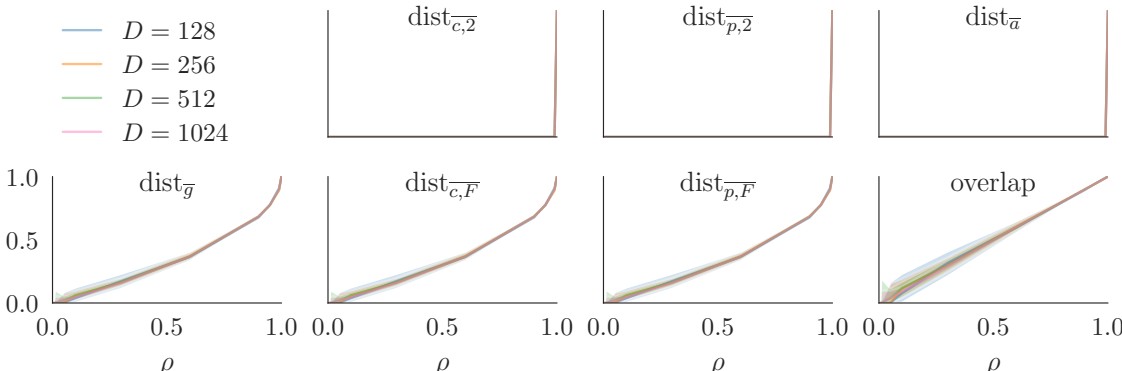

**Figure 11: Grassmannian metrics for random pairs of masks $(Q_1, Q_2) \overset{\text{unif.}}{\sim} (\mathbb{M}^{D \times k}, \mathbb{M}^{D \times k})$ as a function of $\rho$ and for fixed $D$.** Each subplot shows a different Grassmannian metric with the different lines indicating four different dimensions $D$. For each value of $\rho$, we report the median metric over 50 random pairs with the shaded regions showing the 5-95 percentiles.

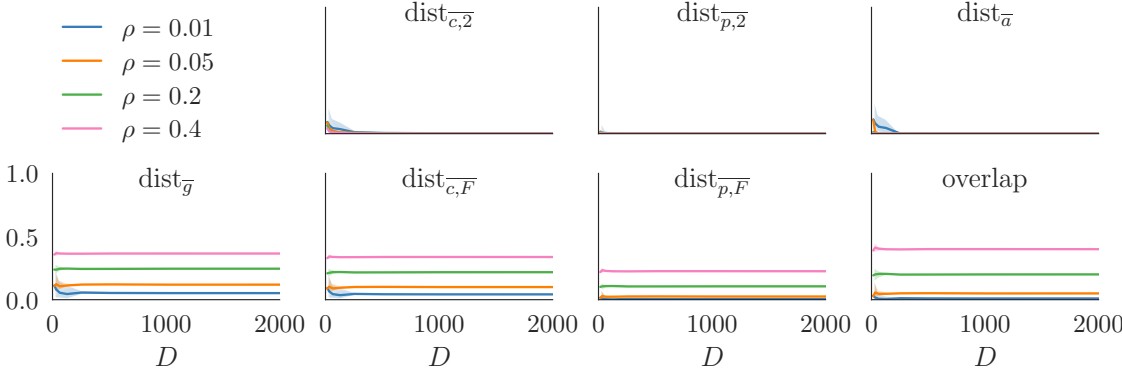

**Figure 12: Grassmannian metrics for random pairs of matrices and masks $(Q_1, Q_2) \overset{\text{unif.}}{\sim} (\mathbb{O}^{D \times k}, \mathbb{M}^{D \times k})$ as a function of $D$ and for fixed $\rho$.** Each subplot shows a different Grassmannian metric with the different lines indicating four different ratios $\rho$. For each value of $D$, we report the median metric over 50 random pairs with the shaded regions showing the 5-95 percentiles.

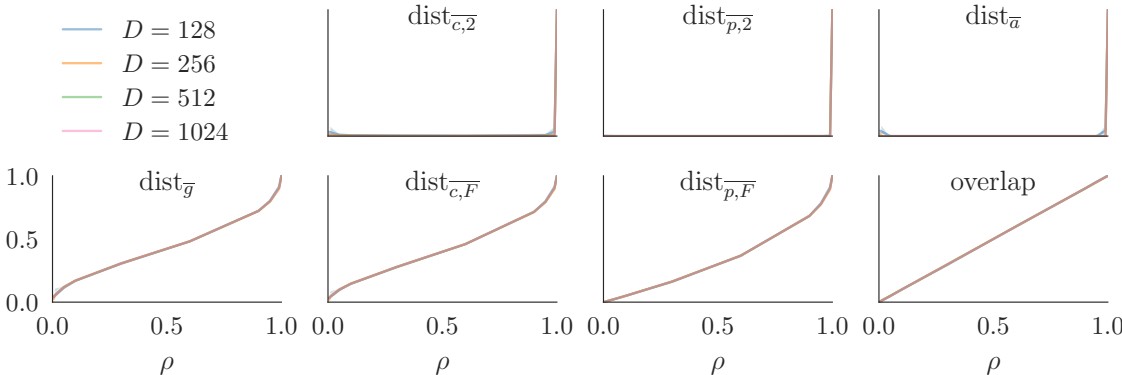

**Figure 13: Grassmannian metrics for random pairs of matrices and masks $(Q_1, Q_2) \overset{\text{unif.}}{\sim} (\mathbb{O}^{D \times k}, \mathbb{M}^{D \times k})$ as a function of $\rho$ and for fixed $D$.** Each subplot shows a different Grassmannian metric with the different lines indicating four different dimensions $D$. For each value of $\rho$, we report the median metric over 50 random pairs with the shaded regions showing the 5-95 percentiles.

### A.7 Analytical *overlap* Baseline vs. Sampling Random DL Masks

In Section 4.2 we justify the use of $\rho := {}^k/_D$ as an analytical, "chance-level" baseline for *overlap*, that we use in Section 6 to observe that the overlap between spaces spanned by magnitude pruning masks and top Hessian eigenspaces in DL is substantially large (see e.g. Figures 3 and 4).

In this context, one natural question one may ask is: *Do random DL masks also lead to higher-than-baseline overlaps?* To answer this question, we trained the $16 \times 16$ MNIST setup following Appendix A.2, and also for steps $t \in \{0, 100, 1000\}$, we computed the *overlap* between the top-$k$ Hessian eigenspace and 10 randomly chosen masks of $k$ nonzeros (the steps were also chosen to cover initialization, mid-training and convergence, as illustrated in Figure 5). Results are shown in Figure 14: We observe that, indeed, *random DL masks tend to behave like our analytical baseline*, which supports the use of the analytical baseline in our experiments.

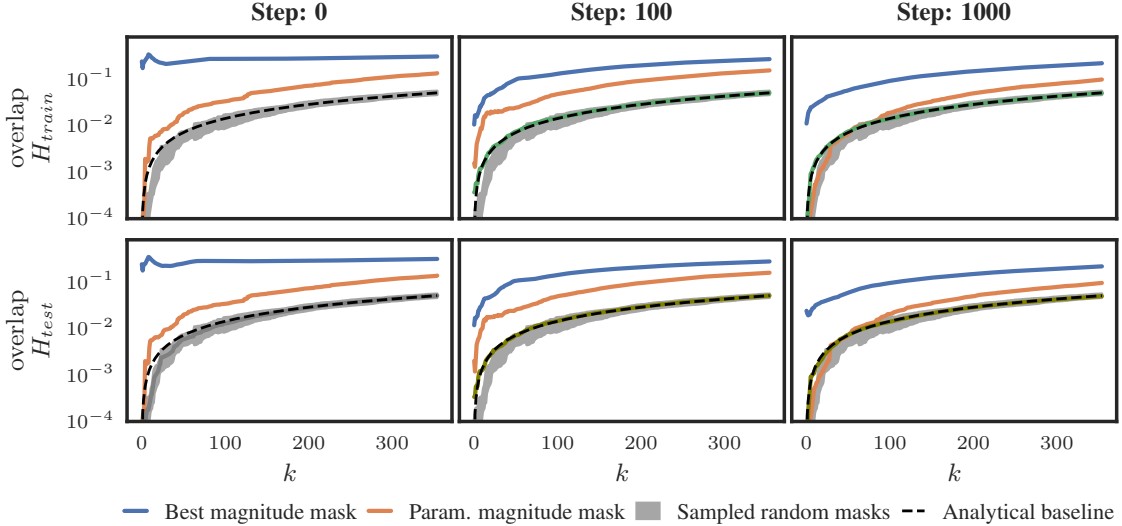

**Figure 14:** *overlap* **measurements for different masking criteria against $H_{train}$ (top row) and $H_{test}$ (bottom row)**. *Best magnitude mask* represents the *overlap* obtained if we pick the $k$ parameters with largest overlap with the top-$k$ eigenspace. *Param. magnitude mask* corresponds to the $k$-largest parameters by magnitude. The *random mask* grey area covers $\pm$ one standard deviation for the *overlap* between $\boldsymbol{H}$ and 10 randomly sampled masks. See Figure 5 for respective accuracies at given steps.

Another natural follow-up question is: *How special are the parameter magnitude masks? Would some other non-random mask also achieve a high overlap?* To tackle this question, we also plot in Figure 14 the *overlap* for the top-$k$ parameter magnitude masks, and the maximal *overlap* achievable by the best possible $k$-mask, serving as upper bound. We observe that, *while the magnitude pruning mask is substantially larger than baseline, it is still far from optimal.*

Still, there is a limited number of masks that can yield high *overlap*: Due to the orthogonality of the Hessian eigenbasis, each column of $\tilde{\boldsymbol{U}}^{(k)}$ must have an $\ell_2$ norm of 1. This means that if a given set of parameters has an *overlap* of $\alpha$, the *overlap* for any non-intersecting set of parameters is at most $1 - \alpha$. Intuitively, the *overlap* is a limited resource, and therefore, masks with an *overlap* significantly above chance-level are bound to be scarce.

## B Supplementary Material for the Experiments in Section 6

### B.1 Results for $16 \times 16$ MNIST

This section includes supplementary material as part of the exhaustive experiments done for the MNIST toy problem. Figure 15 and Figure 16 verify the existence of early collapse and stabilization in both Hessian top space and parameter masks. They also show that SGD was able to successfully train the model under this

setting. Figure 17 corroborates that the behaviour of the Grassmannian metrics analyzed in Section 4 is also present for DL problems.

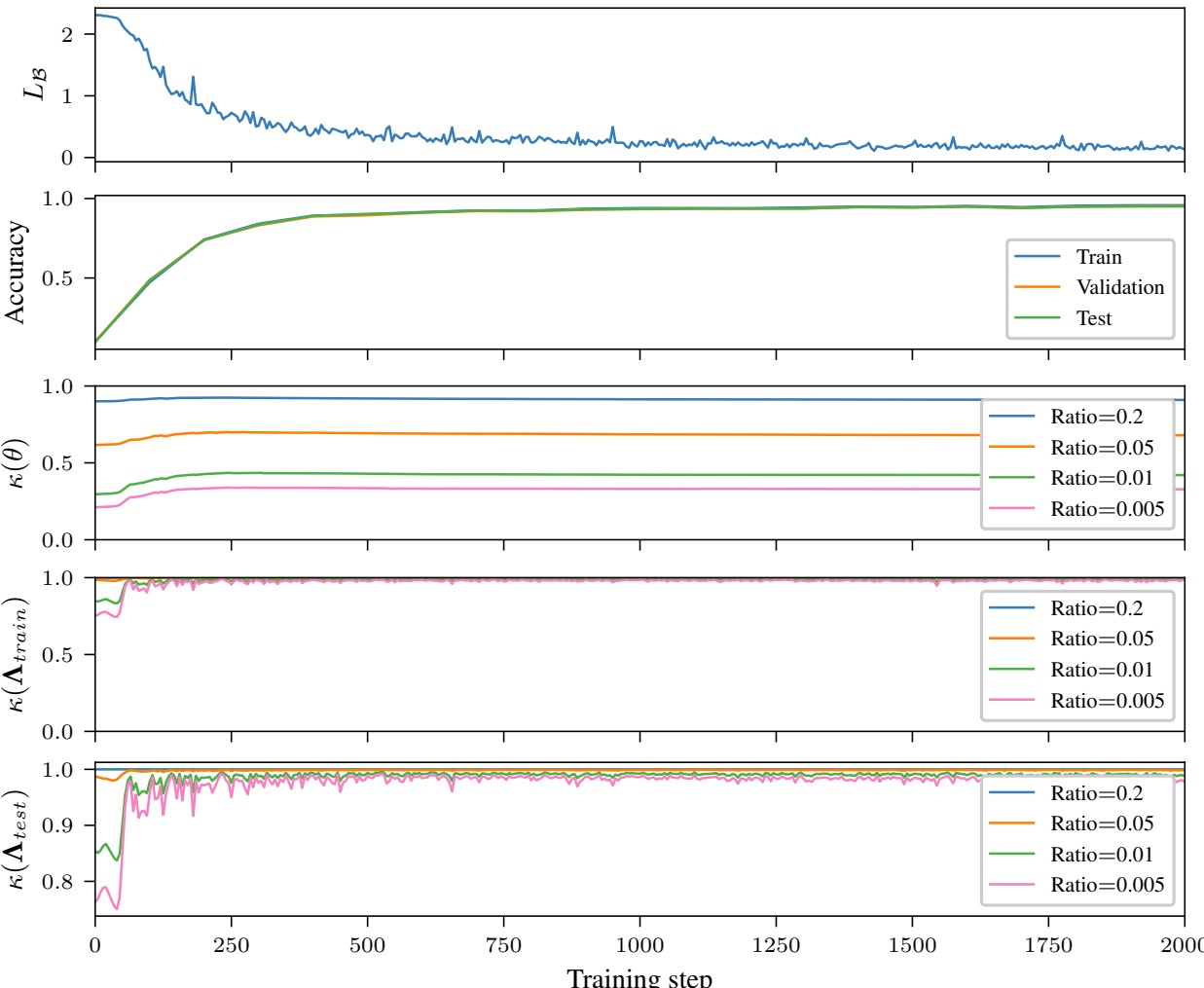

**Figure 15: Both the parameters of the a neural network as well as their Hessian spectrum collapse early during training.** *(top)* The top two subplots show the mini-batch loss $L_{\mathcal{B}}$ as well as the train/validation/test accuracy of the 7030-parameter model trained on $16 \times 16$ downsampled MNIST (see Section 6.1). *(middle)* Looking at the top $20\%, 5\%, 1\%, 0.5\%$ of parameters by magnitude, we can see that very early during training, most of the energy is concentrated on a small subset of the parameters (see Section 2.2 for a definition of $\kappa$). For example, shortly after initialization, the top $0.5\%$ largest parameters by magnitude have roughly $^1/_4$ of the total $\ell_2$ norm of all parameters. *(bottom)* We can observe a similar behaviour for the Hessian spectrum on both the training set (fourth subplot) and the test set (fifth subplot). Only a few steps after training, most of the energy is concentrated in only $0.5\%$ of the eigenvalues.

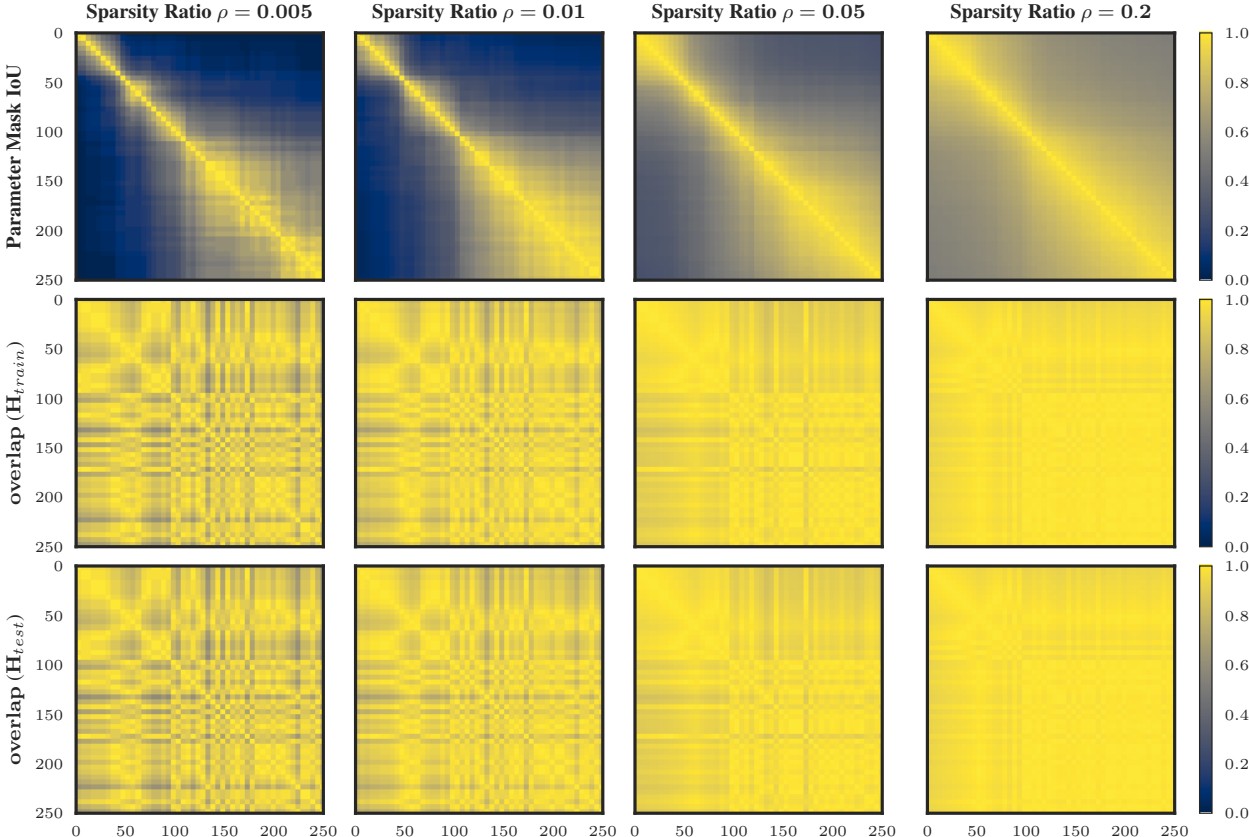

**Figure 16: Both parameter pruning masks (top row) and Hessians (bottom two rows) remain relatively stable after an initial training phase.** Following You et al. (2020), the depicted matrices represent all pairwise similarities (i.e. higher is better) from the beginning of training (top left corners) until step 250 (bottom right corners), for the 7030-parameter model trained on $16 \times 16$ downsampled MNIST (see Section 6.1). For this reason, all matrices are symmetric and have unit diagonals. *(top)* Even when selecting only 0.5% of the parameters (left column), masks collected at different training steps show a remarkable similarity after an initial phase of training. *(middle and bottom)* The *overlap* metric for top Hessian eigenspaces on the train (middle) and test set (bottom) extracted at different training steps and for different subspace sizes (columns). Starting from initialization, the Hessian eigenspaces do not change significantly over the course of training. No substantial differences in behaviour between $\boldsymbol{H}_{test}$ and $\boldsymbol{H}_{train}$ can be observed.

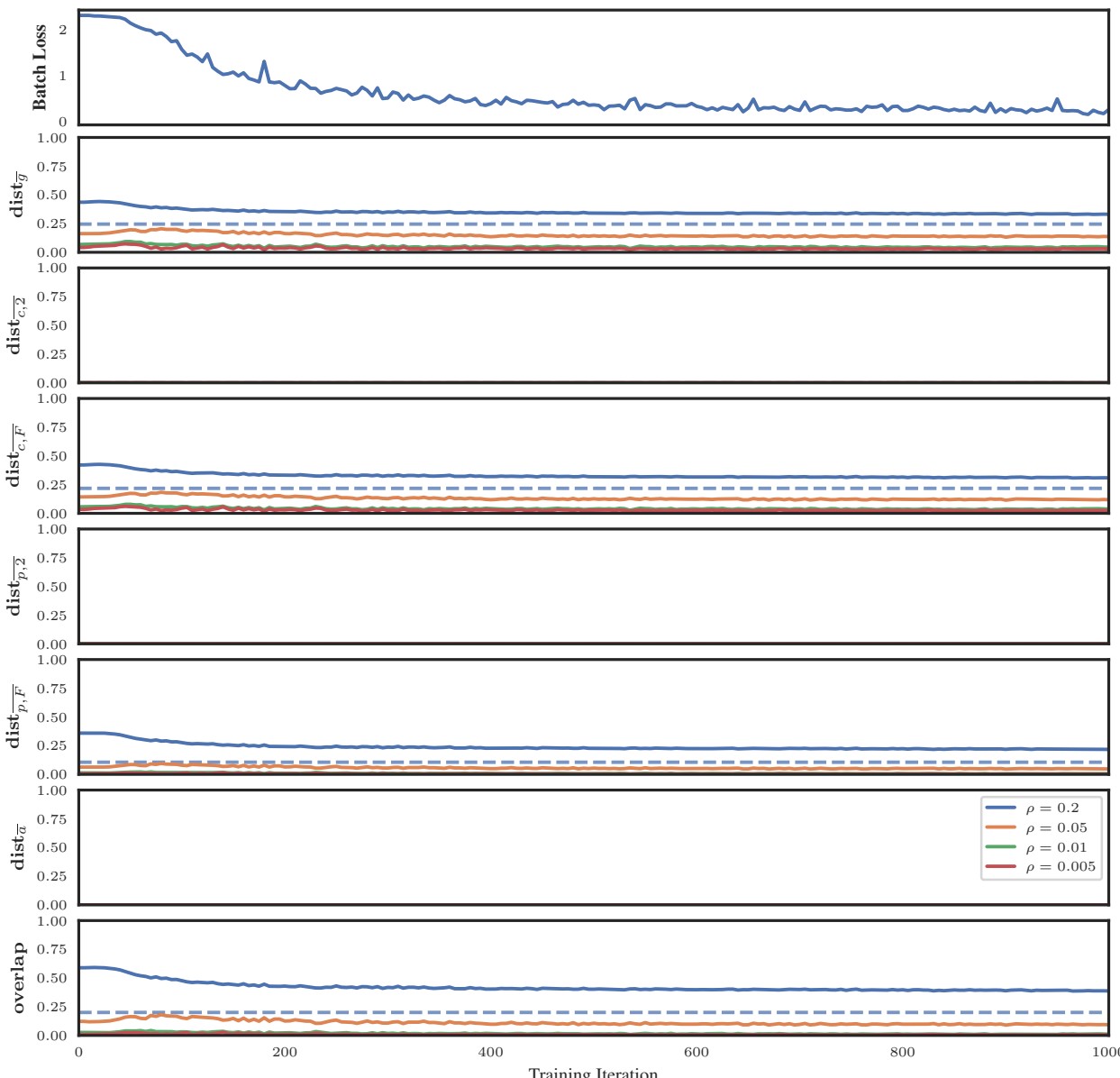

**Figure 17: Different Grassmannian metrics between pruning masks and top Hessian subspaces.** Each line shows a particular sparsity level $\rho$, i.e.. the ratio of unpruned parameters or the size of the top Hessian subspace relative to the full Hessian, as a function of training progress for the 7030-parameter model trained on $16 \times 16$ downsampled MNIST (see Section 6.1). All non-collapsing metrics reveal a consistent and substantial similarity between spaces spanned by pruning masks and top Hessian subspaces well above random chance (random baselines gathered from our synthetic experiments are shown in dashed lines for $\rho = 0.2$), while collapsing metrics are effectively zero due to the large $D$. For better visibility, we limit the plot to the first 1000 steps but all metrics remain stable afterwards.

## B.2 Discussion on Computational Resources

We now discuss the resources involved in the computation of the Hessian eigendecompositions, as well as the experimental design choices surrounding them. The goal is to reflect the scale of computations involved in our experiments, and to convey an idea of the overall resources needed to reproduce them. For more rigorous asymptotics involving memory, arithmetic and measurement costs, readers are invited to review the references exhaustively provided in Section 5, particularly (Tropp et al., 2019, Sec. 4).

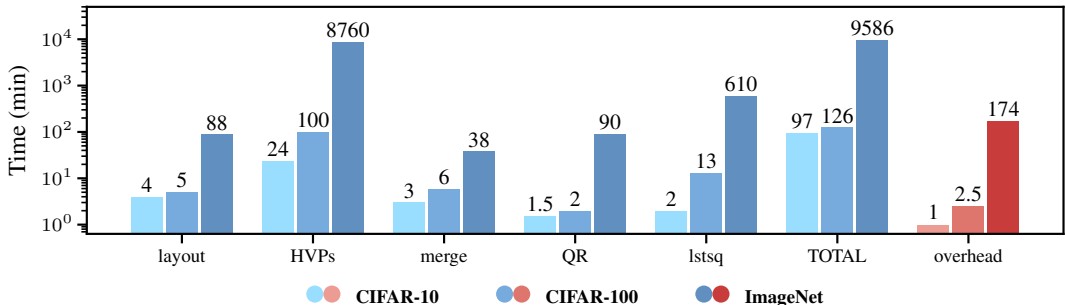

**Figure 18: Runtimes of main SEIGH operations to compute a single Hessian eigendecomposition**, assuming a single computer with 400GB RAM equipped with an NVIDIA A100 (40GB) graphics card. Note that the three involved eigendecompositions feature different networks, datasets, and rank, so a blind one-to-one comparison of runtimes is not possible: refer to Appendix B.2 for an interpretation of this figure.

Figure 18 gathers the average times it took to compute a single sketched Eigendecomposition, for each one of the Hessians covered in Section 6.2. Specifically:

- Resources needed to tune or train the model are here ignored. We used `PyTorch` Paszke et al. (2019).

- The *layout* step allocates the $\mathcal{O}(Dk)$ storage memory in the form of virtual `HDF5` datasets[6]. This allows to efficiently write numerical data in parallel, which is instrumental for the scalability of the procedure.

- The HVPs, computed with the help of `CurvLinOps`[7], correspond to the measurements in lines 1-2 of Alg. 2. As we can see, they take the bulk of the runtime: each HVP requires two forward and backward passes over the Hessian dataset (consisting of 500, 1000 and 5000 samples for the three respective problems, see $N_{train}/N_{test}$ in Table 1). While this can be done in batched fashion, the HVP runtime is linearly affected by dataset size. Another linear factor is the number of measurements (see $n_o$ in Table 1). Crucially, those can be fully parallelized, so the runtimes reported under the *HVPs* column can be divided by the number of available machines. This property of sketched methods, combined with the HDF5 technology, is the main enabler for scalabiltiy here.

- The *merge* step stems from a technicality: most operative systems don't allow a process to open thousands of files at the same time. Hence, we need to merge the virtual HDF5 dataset into a monolithic file. This can be done with only $\mathcal{O}(D)$ memory overhead.

- Although (Tropp et al., 2019, Sec. 4) points out that the *QR decomposition* (line 3 of Alg. 2) dominates the (asymptotic) arithmetic cost for the whole procedure, interestingly this a relatively fast step. We also note that the main reason why we needed so much RAM memory was the need to load $\mathcal{O}(Dk)$ entries in-memory to perform this step. Out-of-core QR orthogonalizations would greatly help reducing RAM requirements.

- The remaining operations (lines 4-8 in Alg. 2) are here gathered under the *lstsq* label, which takes a considerable portion of the overall runtime. To improve this, lines 5 and 6 could be parallelized, roughly halving the runtime, but still faster solutions would greatly help towards more scalable Hessian eigendecompositions.

---

[6]https://github.com/HDFGroup/hdf5
[7]https://github.com/f-dangel/curvlinops

Figure 18 also reports the *TOTAL* runtime of the steps purely involved with the eigendecomposition. Further *overheads*, related e.g. to the use of a computational cluster, are reported separately. In practice, computing e.g. a single RESNET-18 Hessian eigendecomposition with up to fifteen A100s took a bit over a day. While this may seem like a heavyweight computation, it is still a substantial improvement, enabling previously intractable computations. In this section we have also outlined a few ways in which this could be further improved.

Due to the scale of the experiments, we had to be conservative with the choice of hyperparameters. We chose dataset sizes $N_{train}/N_{test}$ such that all classes are balanced, and each class has at least 5 examples. While the chosen number of measurements $n_o$ is much smaller than the ambient dimension $D$, the tremendously rank-defficient structure of $\boldsymbol{H}$ (see Section 2.1) allows for such a disparity. Still, it has been noted that the numerical rank of $\boldsymbol{H}$ may be linked to the number of classes (eg Gur-Ari et al., 2018; Papyan, 2020; Dangel et al., 2022). For this reason we chose $n_o$ values well above this limit.

