# OpenReview forum: "Connecting Parameter Magnitudes and Hessian Eigenspaces at Scale using Sketched Methods"
_TMLR — Accepted by TMLR_

### Review · Reviewer_MZ4v · 2025-01-21

**Summary Of Contributions:**

This paper explores the relationship between parameter magnitude masks and top Hessian eigenspaces in deep neural networks, both of which exhibit early crystallization during training. The authors introduce a methodology to quantify the similarity between these two structures using Grassmannian metrics, identifying overlap as the most interpretable metric. They further propose a scalable, matrix-free eigendecomposition algorithm, enabling analysis of Hessians at unprecedented scales. Empirical results demonstrate a significant overlap between large parameter magnitudes and top Hessian eigenspaces, highlighting their structural connection. This work has implications for optimization, pruning, and uncertainty estimation in deep learning.

**Audience:**

Yes

**Claims And Evidence:**

Yes

**Requested Changes:**

To improve practical relevance, it would be helpful to provide examples or experiments showing how the discovered connection can be leveraged to optimize training, enhance pruning techniques, or improve uncertainty estimation.

**Strengths And Weaknesses:**

Strengths:

- Establishing a direct link between parameter magnitudes and Hessian eigenspaces provides a novel perspective on neural network structure and optimization.
- The development of a sketched eigendecomposition algorithm addresses the computational challenges of analyzing large-scale Hessians, enabling experiments with models containing over 10 million parameters.
- The consistent and substantial overlap observed across various models and datasets reinforces the proposed connection, with effects becoming more pronounced for larger networks.
- The findings are applicable to several areas, including pruning, optimization, and understanding loss landscapes, making the contributions broadly impactful.


Weaknesses:

- The analysis primarily emphasizes early crystallization and does not explore whether these overlaps persist or evolve in later training stages.
- Despite scalability improvements, the resource requirements for large-scale experiments seem to be substantial, potentially limiting adoption.
- While the theoretical insights are significant, the paper does not directly demonstrate how these findings could be applied to improve training efficiency or model performance in real-world settings.

---

> ### Author Response · Authors · 2025-02-14
> **Thank you for your review**
>
> Dear Reviewer MZ4v,
> Thank you very much for your thoughtful review, which we acknowledge.
> We will be happy to address your comments once we have received all reviews.
> Thank you for your patience.

---

> ### Author Response · Authors · 2025-03-04
> **Answer to Reviewer `MZ4v`**
>
> Dear Reviewer `MZ4v`,
>
> thank you very much for reviewing our paper and providing valuable feedback.
> We also appreciate your remarks on the broad impact of this work, in terms of scalability and potential downstream applications.
> We address each of your comments separately below, and have updated our manuscript accordingly.
> Please let us know if any further clarifications are needed.
>
>
> ## Analysis emphasizes early crystallization
>
> Thank you for the opportunity of clarifying this:
> Indeed, our motivation to study parameter and curvature spaces jointly is the early crystallization they exhibit, but our developed methodology works at any point during training.
> This is in contrast with previous work, which has explored this connection at convergence and for individual parameters, as we emphasize in the last paragraph of the introduction.
>
> As for our findings, it is true that we report overlap only until step 8000. We did so to allow for easier comparison among problems, since the observed phenomenon is clearest in that range. Still, this analysis is not limited to the early stages of training: As it can be seen in Table 1 (section 6), the accuracy obtained at reported maximal steps is up to 95.78% for MNIST, which is arguably beyond early-stage.
>
> We have updated the first paragrah of section 6 emphasizing this aspect, which we agree is relevant.
>
>
> ## Scalability issues:
>
> This is rightfully pointed out: computing Hessians at scale is a crucial problem and can limit adoption of Hessian-based research/methods.
> We believe that one relevant contribution of our research is to enable this on previously infeasible scales.
> However, this doesn't mean that all work is done: we agree that there is still room for improvement in terms of scalability, perhaps via different metrics or other Hessian approximations, and that this may limit adoption.
> We have added a new section to the Appendix, where we propose and study three different perturbation-related measurements that could be used as scalable proxies for overlap.
>
> Still, we note that measuring the overlap is meant to be an *analysis* technique to understand the connection between parameters and loss curvature, and not necessarily something that needs to be computed when performing a downstream application.
> We have added this remark at the end of the conclusion section.
>
>
> ## Unclear how to leverage in downstream applications (pruning, training, uncertainty estimation):
>
> We agree that much work is still to be done in order to explore and leverage the connection between parameters and loss curvature.
> In this context, not only efforts in developing novel methods, but also in explaining existing ones (magnitude pruning, natural gradient descent/momentum, last-layer Laplace...) are of key interest.
> We believe that our work paves the way for such efforts, by establishing a methodology to measure said connection in a stable, scalable and theoretically sound manner.

---

### Review · Reviewer_bbvJ · 2025-02-07

**Summary Of Contributions:**

In this work the authors explore the connection between large parameter magnitudes, which are used for sparsity masking, and the structure of the Hessian. They use a sketching method to show that indeed there is high overlap between parameters with large magnitudes and the large eigenspaces of the Hessian in a variety of models. This suggests a strong link between popular sparsification methods and loss landscape geometry.

**Audience:**

Yes

**Claims And Evidence:**

Yes

**Requested Changes:**

It is argued in Appendix A.1 that a perturbation approach would fail due to the non-PSD nature of the Hessian. There are a couple of paths around this: first is to use the Gauss-Newton component of the Hessian. This is PSD, contains most of the information about large eigenspaces, and GN-VP are generally easy to compute. Another alternative are metrics related to the squared Hessian, which are also obtainable by HVPs. Note that there are also perturbation-based approaches to obtaining this information without HVPs; see the relationship between Equations 2, 3, and 7 in [1]. By replacing $\epsilon$ in this paper with the desired perturbation (e.g. a onehot vector representing perturbation of a single large parameter), I believe interesting statistics can be recovered.

**I would like to see some experimental results on at least one of the above simpler metrics before acceptance; I do believe that the authors' methodology is sound, but this would be a good tie-in to simpler approaches.**

Figure 2, top row is a bit hard to parse - maybe log y axis there?

The authors invoke the geometry of Grassmanians to define overlap. Overlap can also be motivated more simply via cosine similarity induced by the standard Euclidean metric on matrices, and the observation that the metric is rotationally invariant. Perhaps this can be used as the initial intuition building motivation, and then the Grassmanian formulation presented next in order to solidify the foundation believe this will improve digestibility of the paper by the ML community, as a broad swath of researchers will find this work interesting.

For section 4.2: might be better to start with the statement that overlap is the focus of the rest of the paper, and then go into the explanations for why? I also think that the expressions for the other metrics should be in the main text (or, at the very least, the specific equation numbers from the appendix should be referenced).

In Section 5 the authors criticize Lanczos methods for not generating eigenpairs; the approximate eigenvectors generated by the lanczos process can indeed be used to estimate eigenvectors for large eigenvalues in particular. In my opinion this is an important point.

In the presentation of the sketching algorithms, for Algorithm 2, what are the subscripts on $\Omega$ in steps 1 and 2? It would also be good to indicate the dimensions of the output matrices as well. Additionally, indicating which dimensions are ``big'' vs small will be helpful. I also wonder if in the main text description there is a way to make more clear which quantities are obtained using HVPs strung together. May also be worth mentioning that a $V$-vp is implemented by composing vector-mask product with HVP (if I understand correctly!).

Could be nice to add something like ViT to the experiments -  I think a S/16 or Ti/16 sized model should be pretty easy to run under these methods even with modest compute resources. **This experiment is not necessary to secure publication recommendation**.

[1] https://arxiv.org/abs/2502.02407

**Strengths And Weaknesses:**

The main strength of the paper is the clarity of the question and the approach used to answer it. Linking together sparsification procedures with the geometry of the loss landscape is a highly relevant topic. The sketching approach is a great fit here, and the descriptions of the overlap metric and its connection to sketching are also highly informative.

Main weaknesses are due to presentation and diversity of model/dataset pairs; see **requested changes** below. An additional experiment using Gauss-Newton or squared Hessian metrics is necessary to secure my acceptance; I believe this is a simple experiment to run given the author's setup (requires a modest number of HVPs at checkpoints studied by the authors). Once this experiment is added I will likely switch my claims and evidence to "yes".

Edit: Claims and evidence switched to "yes" after authors made changes to the text, including adding the requested experiments.

---

> ### Author Response · Authors · 2025-02-14
> **Thank you for your review**
>
> Dear Reviewer bbvj,
> Thank you very much for your thoughtful review, which we acknowledge.
> We will be happy to address your comments once we have received all reviews.
> Thank you for your patience.

---

> ### Author Response · Authors · 2025-03-04
> **Answer to Reviewer `bbvJ` (part 1/2)**
>
> Dear Reviewer `bbvJ`,
>
> Thank you very much for your review and your constructive feedback, which we have integrated into an updated version of our manuscript.
> We are also grateful that you recognize the clarity and relevance of our work.
> You can find our answer to your comments below, including the requested experiments.
> Please let us know if any further clarifications are needed.
>
> ## Figure 2, top row is a bit hard to parse
>
> Thank you for pointing this out.
>
> The issue with collapsing metrics like the 2-Chordal norm is that they are numerically zero almost everywhere, except for very low $D$ or very large $\rho$ values. Figure 2 highlights precisely this.
> Due to the abundant zeros, a log y-axis would render NaNs or lines out of bounds almost everywhere, to the same effect as the current plot. Same with a truncated linear axis would still have a similar appearance. All of these alternatives will also come with the downside of losing visual comparability to the other rows.
>
> Alternatively, one could just not include any collapsing metric, but this would defeat one main purpose of this figure.
>
> For this reason, we believed that a better way to help clarifying this issue is to explaining via text that this metric is numerically zero everywhere, as seen in the caption of Fig.2. This behaviour is also mentioned in: Fig. 14, Section 4.2, Appendices A.3/A.6.
> We are happy to consider further suggestions on how we can make Figure 2 more readable and informative.
>
> ## Overlap can also be motivated more simply via cosine similarity
>
> Indeed, that is one valid (and perhaps more accessible) way of regarding overlap.
> We agree clarifying this will be helpful to make our work more digestible and impactful, thank you for this insight and for the words of support in that respect.
> We updated the beginning of section 4 including this insight.
>
>
> ## Clarity and comprehensiveness of Section 4
>
> Thank you for suggesting these changes to our Grassmannian section, which we agree will greatly improve clarity.
> We have incorporated the metric definitions from the appendix into Section 4.1 (note that this is about half a page, still well within the boundaries of the publication format).
> We believe that this, together with the focus and emphasis on overlap in the previous subsection, helps improving clarity and comprehensiveness of section 4.
>
>
> ## Lanczos eigenvectors
>
> Thank you for pointing out this mistake.
> It is indeed true that Lanczos iterations can be used to generate eigenpairs (typically associated with extremal eigenvalues), via the so-called Ritz vectors.
> Our formulation in section 5 made it sound like this is not the case. We agree that it is important to rectify this: we updated the corresponding paragraph clarifying this matter, and also elaborating a bit on this aspect.
>
> In our update, and also here, we want to remark that the main point of the section, which goes along section 1.4.2 of [Halko et al., 2011], still holds: sketched decompositions have the advantages of being inherently stable even for larger $k$, and parallelizable, which is crucial when measurements take a long time, as is the case with HVPs (this is also highlighted in Appendix B.2).
>
> # Clarity of Algorithm 2
>
>
> > In the presentation of the sketching algorithms, for Algorithm 2, what are the subscripts on in steps 1 and 2?
>
> Indeed, the abused subindex notation for $\Omega$ lacks clarity.
> We meant column vectors of $\Omega$, but we have reformulated the algorithm in terms of matrix concatenations, and all quantities and operations are now defined. Thank you for pointing this out.
>
> > It would also be good to indicate the dimensions of the output matrices as well. Additionally, indicating which dimensions are ``big'' vs small will be helpful.
>
> We agree this would help understanding the involved objects and computations.
> We added a sentence to section 5 about the big vs. small dimensions involved in the SSVD.
> We have also added the dimensions of the output matrices to both SSVD and SEIGH algorithms.
>
> > I also wonder if in the main text description there is a way to make more clear which quantities are obtained using HVPs strung together.
>
> Thank you for this suggestion: this is a crucial computation that deserves more emphasis.
> We added a sentence to the SSVD explanation, after the $N_I + 2 N_O$ measurements are mentioned, explaining which measurements are used to obtain which objects, and pointing to the corresponding lines in the algorithms.
>
> > May also be worth mentioning that a V-vp is implemented by composing vector-mask product with HVP (if I understand correctly)
>
> We are not sure we understand this remark. The sketched algorithms introduced perform plain HVPs using random measurement vectors that are not masked. Masking happens when computing the overlap against a particular parameter subset.
> Perhaps this lack of clarity will be resolved once we put more emphasis on the measurement step.
> Otherwise, feel free to elaborate on this aspect for further discussion.

---

> > ### Comment · Reviewer_bbvJ · 2025-03-04
> > **Thanks for the edits**
> >
> > Thank you for the changes! The paper is much improved.
> >
> > The point about Figure 2 is a reasonable one. I *think* that using a `symlog` scale with `linthresh = 1e-3` may improve the clarity of the figure significantly, while highlighting your points.
> >
> > In Section 5, the paragraph titled "A case for sketched decompositions:" is a bit dense - it might be easier to read if there were subparagraphs for each method discussed?
> >
> > The additional reminders of which dimensions are large or small are helpful. I wonder if this could be made even better with notation: maybe using a lower case letter to indicate dimensions which are small, and upper case for those that are large? Just a thought.
> >
> > >May also be worth mentioning that a V-vp is implemented by composing vector-mask product with HVP (if I understand correctly)
> >
> > Ignore this comment, it was due to a misunderstanding on my part.
> >
> > I am updating claims and evidence to "yes" in light of the changes.

---

> > ### Comment · Reviewer_bbvJ · 2025-03-04
> > **Another thought**
> >
> > It might be good to refer to the results of Appendix A.2 in the conclusion as well - readers may miss these alternative methods on first pass, and I think the experiments are quite nice!

---

> > > ### Author Response · Authors · 2025-03-04
> > > **Response to additional thoughts**
> > >
> > > We are glad that you also consider the extended perturbation results a good addition to the paper.
> > >
> > > Also thank you very much for the prompt response, and for the additional suggestions and thoughts. We agree that they can further improve clarity and impact. We will explore incorporating them to the manuscript as soon as possible.

---

> > > > ### Author Response · Authors · 2025-03-19
> > > > **Incorporated some of the additional thoughts**
> > > >
> > > > Thanks again for the helpful suggestions. We have updated the manuscript with the following changes:
> > > >
> > > > * Small- vs big-dimensional notation: now the measurement dimensions (small) are lowcase: $n_o, n_i$ instead of $N_O, N_I$. Also added a footnote to page 7 explaining that lower/uppercase reflects dimension size.
> > > > * Added pointer to Appendix A.2 in the conclusion (note that we had already added a pointer in the beginning of Section 4).
> > > > * Fixed typo: last paragraph of Section 5, $M_I$ is actually $N_I$.

---

> ### Author Response · Authors · 2025-03-04
> **Answer to Reviewer `bbvJ` (part 2/2)**
>
> ## Suggested approaches for perturbation study
>
> In the main body we argue that the Grassmannian methodology, in and of itself, is a valid way of measuring the relation between parameters and curvature throughout training.
> Still, we sympathize with the interest in finding scalable alternatives, also as a "tie-in" to simpler approaches, in the spirit of e.g. the experiment we performed in Appendix A.7.
> We have performed the 3 suggested experiments, which we added to a new appendix section A.2.
>
> We appreciate the suggested approaches, in their variety and potential to achieve these goals. Based on a closer analysis, we made the following interpretations, which may deviate slightly from the suggestions, but we believe retain the spirit more truly. Specifically:
> * For the Gauss-Newton and squared Hessian, we directly measure the traces, as it is comparably efficient and removes the $R_3$ source of error.
> * For the SAM perturbations, we do not look into the GGN decomposition from ( https://arxiv.org/abs/2502.02407 ), since PSD-ness is anyway not necessary when performing one-hot masking (as argued in Appendix A.1).
> * To ensure comparability with overlap, we propose features and a baseline based on normalized sums
>
> This resulted in 3 more scalable approaches with interesting, albeit mixed results, reported in Appendix A.2.
> We are grateful for these suggested approaches, which we believe have substantially enriched the paper.
> If you agree, we would appreciate if you reconsider accepting our paper for publication.

---

### Review · Reviewer_iShm · 2025-02-18

**Summary Of Contributions:**

The paper focuses on the observed connection between the sparsification of the deep network early in training and the stabilization of the top eigenspace of the Hessian. The methodology is to measure the alignment of the top Hessian subspace to the coordinates of the biggest magnitude parameters using $ \frac{1}{k} || I^T S ||_F^2$  where $S$ is the top subspace, that is the so-called (k) overlap.
A sketched SVD method is proposed and validated to compute $S$ which scales well even for Image-net scale architectures. The measured overlap is above the chance level supporting the observed connection between sparsification and eigenspace stabilization.

**Audience:**

Yes

**Broader Impact Concerns:**

None.

**Claims And Evidence:**

Yes

**Requested Changes:**

None.

**Strengths And Weaknesses:**

**Strengths**

- The paper is very well written
- Experiments are clean and scale well up to Image-net
- Overall, I think it is a nice contribution supporting the alignment between the surviving parameters of the deep net with the top eigenspace of the Hessian, not only at convergence but early in training
- The alignment is much more significant for larger networks, suggesting that this effect could be even provable for some infinite-width/depth architectures

**Weaknesses**

*Not really weaknesses two questions:*

- Fig 13 shows both the largest magnitude parameters and the top eigenspace stabilizes early in training. I wonder how much stronger (if it is stronger) this self-stabilization effect is compared to the alignment between the largest magnitude param. space and the top eigenspace (?) A future investigation could be if the stabilization of the largest param. subspaces imply the stabilization of the top Hessian eigenspace.

- I wonder if the authors tried this to measure the alignment $\frac{1}{k} \max_P \text{Tr} (I^T P S )$ where $P$ is a $k \times k$ permutation matrix. This would first match $k$ (orthonormal) vectors to other $k$ orthonormal vectors and then sum up over $k$ angles between the matched pairs. It is smaller than the overlap selected in the paper as it removed the angles between the vectors that are not closest to each other. I'm not sure how computationally expensive finding $P$ would be but if it is not expensive this could be a more stable measure (?)


-------------------------------------------------------------------------------------------------------------
PS. I apogize for the delay in review. I got stuck outside of my residence country due to visa issues.

---

> ### Author Response · Authors · 2025-03-04
> **Answer to Reviewer `iShm`**
>
> Dear Reviewer `iShm`,
>
> Thank you very much for your thoughtful review.
> We are happy that you recognize the usefulness and significance of our findings.
> We address your questions below, please let us know if any further clarifications are needed.
>
> ## Relation between overlap and stabilization
>
> Indeed, our work uses the early crystallization of parameters and curvature just as a motivation to study their relationship, and then our focus shifts to establish the similarity of their spanned subspaces at scale.
>
> But we agree that there is much more to be investigated beyond the similarity of their spanned subspaces, including explaining why would such similarity arise and whether one crystallization causes the other.
> A possible idea worth exploring in future work, speculatively, would be to apply the Schur-Horn theorem to obtain bounds for the Hessian eigenvalues, given the Hessian diagonal entries.
>
> ## Proposed stable measure
>
> In our work, we propose *overlap* due to its stability and Grassmannian characterization, but we acknowledge that there are a variety of alternative metrics that could be used.
> In particular, in Appendix A.7 we show that the permutation matrix $P$ that emerges from magnitude masks is far from optimal, and we agree that this behaviour merits further study.
> Furthermore, we have added a new section to the Appendix, where we propose and study three different perturbation-related measurements that could be used as scalable proxies for overlap.
>
> As for your proposed metric, we are not sure we fully understand it. if $S$ is a top subspace of shape $D$ by $k$, would not $P$ be a $D$ by $D$ matrix? This would make the identity redundant. Perhaps you are suggesting to find the $k$ rows from the top subspace with maximal norm (trace would not qualify since eigenvector entries can generally be positive or negative). This is precisely the blue line in Figure 11 (Appendix A.7), and would correspond to a top-$k$ eigenspace mask", as opposed to parameter magnitude (orange). This is indeed a way of obtaining masks that merits further study, which we hope to enable also with our scalable method.

---

### Decision · Action_Editor_cKaj · 2025-03-21

**Recommendation:** Accept as is

**Comment:**

All reviewers agree that the paper is well-written, clear, and provides interesting insights that are of interest to the TMLR community. There were some questions raised by reviewers on, for example, using simpler measures of overlap via cosine similarity and additional experiments like perturbation study, but the authors did a good job in responding to these questions, and at the end of the rebuttal period, all reviewers unanimously recommended acceptance of the paper and AE concurs with this judgment and recommends accept as-is. However, the authors are still recommended to think more deeply about some open questions raised by the reviewers, such as the practical implications of their work and some additional, qualitative discussions outlining concrete use cases will be beneficial to further increase the impact of their work.

**Audience:**

The paper connects two well-studied areas and provides novel insights and are of interest to TMLR audience.

**Claims And Evidence:**

This paper makes a connection between network sparsification by weight magnitude and the observation that during neural network training, the majority of the loss landscape's curvature concentrates in a tiny eigenspace of the loss Hessian. The paper then proposes an overlap metric that measures the similarity between parameter masks and Hessian eigenspaces, and discovers that the overlap between magnitude parameter masks and top Hessian eigenspace is higher than chance-level and this effect holds true even for large, ImageNet-scale networks.

Overall, the paper makes an interesting connection in two areas that are respectively well-studied but rarely studied together, and all reviewers agree that the experiment is comprehensive and the findings support the claim, it is particularly commendable that the authors show their claim scale to large-scale networks. The AE concurs with this view and believes that the paper meets the "Claims and Evidence" bar.